# Facile syntheses of conjugated polymers for photothermal tumour therapy

Peiyao Chen[1], Yinchu Ma [2], Zhen Zheng[1], Chengfan Wu[1], Yucai Wang[2] & Gaolin Liang [1,3]

Development of photothermal materials which are able to harness sunlight and convert it to thermal energy seems attractive. Besides carbon-based nanomaterials, conjugated polymers are emerging promising photothermal materials but their facile syntheses remain challenging. In this work, by modification of a CBT-Cys click condensation reaction and rational design of the starting materials, we facilely synthesize conjugated polymers poly-2-phenyl-benzobisthiazole (PPBBT) and its dihexyl derivative with good photothermal properties. Under the irradiation of either sunlight-mimicking Xe light or near-infrared laser, we verify that PPBBT has comparable photothermal heating-up speed to that of star material single-wall carbon nanotube. Moreover, PPBBT is used to fabricate water-soluble $NP_{PPBBT}$ nanoparticles which maintain excellent photothermal properties in vitro and photothermal therapy effect on the tumours exposed to laser irradiation. We envision that our synthetic method provides a facile approach to fabricate conjugated polymers for more promising applications in biomedicine or photovoltaics in the near future.

[1] Hefei National Laboratory of Physical Sciences at Microscale, Department of Chemistry, University of Science and Technology of China, 96 Jinzhai Road, 230026 Hefei, Anhui, China. [2] The CAS Key Laboratory of Innate Immunity and Chronic Disease, School of Life Sciences and Medical Center, University of Science and Technology of China, 230027 Hefei, Anhui, China. [3] State Key Laboratory of Bioelectronics, School of Biological Sciences and Medical Engineering, Southeast University, 210096 Nanjing, Jiangsu, China. These authors contributed equally: Peiyao Chen, Yinchu Ma, Zhen Zheng. Correspondence and requests for materials should be addressed to Y.W. (email: yucaiwang@ustc.edu.cn) or to G.L. (email: gliang@ustc.edu.cn)

Energy is indispensible for industrial production and daily life. Currently the major sources of energy are fossil fuels. Combustible fossil fuels combust in heat engines to generate mechanical energy but in the meantime bring severe pollution problems[1]. Therefore, in terms of energy exploitation and environment protection, developing photothermal materials, which are able to harness sunlight and convert it to thermal energy seems extraodinarily attractive[2–4]. Among the various types of photothermal materials, carbon-based materials are good absorbers due to their π-band's optical transitions[5]. In particular, carbon nanotubes (CNTs) exhibit extraordinarily high efficiency of photothermal conversion and excellent thermal conductivity $(2000–76,000 \, W \, m^{-1} \, K^{-1})$, which is comparable to that of the best natural thermal conductor diamond[6,7]. Besides carbon-based nanomaterials, conjugated polymers, whose large non-localized π-bond backbones are composed of aromatic or aromatic heterocyclic rings, represent another type of emerging photothermal materials[8,9]. Capable of transducing light into local heat, some conjugated polymers with prominent photothermal conversion efficiency ($\eta$) have been applied as an efficient medical tool in photothermal therapy (PTT) for cancer treatment[10–12]. Accumulating evidences have indicated that nano-agent (e.g. CNT)-assisted PTT might bring anti-tumour immunological effects by generating tumour-associate agents from ablated tumour cell residues[13], rendering PTT a powerful tool for cancer therapy. Nevertheless, facile fabrications of these photothermal materials remain challenging. At present, there are many techniques to fabricate CNT in sizable quantities including arc discharge, laser ablation, high-pressure carbon monoxide disproportionation, and chemical vapor deposition[14]. However, most of these manufacturing processes take place in vacuum or with processing gases. Compared to the harsh conditions for CNT fabrication, the syntheses of conjugated polymers are usually more flexible, solution-processable, or lower-cost[15]. Moreover, some conjugated polymers were reported to interact with the immune system and might be used as potenial vaccine adjuvants[16]. However, current monomer species for the syntheses of conjugated polymers are quite limited and therefore available conjugated polymers with excellent photothermal properties are quite few[17]. This calls for new molecular structures and efficient reactions to fabricate conjugated polymers for the applications of energy exploitation.

In 2010, after carefully investigating the luciferin regeneration pathway in firefly, Rao et al. developed a click condensation reaction between the cyano group of 2-cyano-benzothiazole (CBT) and the 1,2-aminothiol group of cysteine (Cys), which could be controlled by pH, reduction, or protease[18]. This CBT-Cys click condensation reaction has a very high second order reaction rate of $9.19 \, M^{-1} \, s^{-1}$ and is biocompatible to biological systems[19]. To date, this click condensation reaction has been successfully utilized to design smart imaging probes (optical, nuclear, or magnetic resonance probes)[20–23], synthesize cyclic superstructures[24], overcome multidrug resistance[25], and prepare oligomeric hydrogels[26]. Nevertheless, there has been no report of using this click condensation reaction for even the fabrication of conjugated polymeric materials, let alone applications of such type of materials (e.g. photothermal therapy).

Inspired by the abovementioned pioneering studies, in this work, we modify abovementioned CBT-Cys click condensation reaction and rationally design the starting materials for the facile fabrication of conjugated polymers. We hypothesize that, as exampled in Fig. 1a, if two 1,2-aminothiol groups of above click condensation reaction are on one aromatic ring but not cysteines, and two cyano groups are directly linked to one benzene ring but not benzothiazoles, these two starting materials could still be subjected to above click condensation reaction to yield π-conjugated polymers with excellent photothermal

properties. Therefore, 1,4-dicyanobenzene and 2,5-dimercapto-1,4-phenylenediamine are designed as the starting materials to synthesize poly-2-phenyl-benzobisthiazole (PPBBT, $R = H$ in Fig. 1a). The resulting PPBBT exhibit an ultraviolet-visible-near-infrared (UV-Vis-NIR) spectrum in good overlap with solar spectrum, suggesting its potential application in solar energy ultilzation. Using single-wall CNT (SWCNT) as the control, we verify that as-obtained PPBBT has comparable photothermal heating-up speed to that of SWCNT. Moreover, as illustrated in Fig. 1a, via aromatic substitution, PPBBT could be easily modified with additional functional groups, demonstrating the feasibility of our approach in large-scale screening of conjugated polymers with desired properties. Finally, we find that the hydrophobic PPBBT could be easily coated with Poly(ethylene glycol)-block-poly(hexyl ethylene phosphate) (mPEG-b-PHEP) through routine nanoprecipitation method to obtain highly stable, physiologically dispersed, and photostable nanoparticles $NP_{PPBBT}$. Notably, as-prepared $NP_{PPBBT}$ nanoparticles show low dark-cytotoxicity but high photo-cytotoxicity at $50 \, \mu g \, mL^{-1}$. In vivo experiments show that the $NP_{PPBBT}$ nanoparticles are effectively accumulated in the tumour site due to their enhanced permeability and retention (EPR) effect and have excellent PTT efficiency against tumour.

## Results

**Facile syntheses and optical properties of PPBBT.** By modifying abovementioned CBT-Cys click condensation reaction and employing the reported $Cu^{2+}$-catalyzed reaction condition between 2-aminobenzenethiol and benzonitriles[27], we facilely synthesized PPBBT and its derivative Dihexyl-PPBBT (Fig. 1a). Five minutes after equivalent 1,4-dicyanobenzene and 2,5-dimercapto-1,4-phenylenediamine at 67.6 mM were mixed in ethanol in the presence of 10 mol% of Cu(Ac)$_2$ at 70 °C, black precipitates of PPBBT in large quantity (yield: 59.9%) were observed, suggesting the reaction rate of this condensation reaction is very high. The crude product was then centrifuged, washed with water and ethanol to remove the catalyst and remaining starting materials, and lyophilized to yield pure black powder product (Fig. 1b) and characterized with $^1H$ NMR spectrum (Supplementary Figure 1). The successful formation of the poly-benzothiazole structure was characterized with X-ray photoelectron spectroscopy (XPS) spectra in Supplementary Figures 2 and 3. As shown in the Supplementary Figure 2, the newly appeared N $1s$ peak at 398.7 eV of PPBBT could be assigned to the $sp^2$-hybridized nitrogen in the newly formed C–N–C structures on the benzobisthiazole structure. Additionally, element composition analysis of the XPS spectrum suggested that the atom number ratio between N and S (N/S, 1.02) was near to 1:1 (Supplementary Figure 3 and Supplementary Table 1), indicating a very high polymerization degree of the PPBBT product (note, the N/S ratio of the two starting materials was 2:1 while that of the repeating unit in the polymer product is 1:1). Additionally, the gel permeation chromatography (GPC) trace of PPBBT showed that the number-average molecular weight of the polymer was about 23,176 with a polydispersity (PDI) of 1.34 (Supplementary Figure 4 and Supplementary Table 2). To test the feasibility of introducing additional functional groups onto PPBBT, we synthesized 2,5-dihexyl-1,4-dicyanobenzene[28] (Supplementary Figures 5–8) and used it to react with 2,5-dimercapto-1,4-phenylenediamine under the same reaction condition for the preparation of dihexyl-substituted PPBBT (Dihexyl-PPBBT, Fig. 1a). As expected, the black powder Dihexyl-PPBBT was also facilely obtained with a yield of 69.0% (Fig. 1b). Successful formation of the poly-benzobisthiazole structure in Dihexyl-PPBBT was also characterized by its XPS spectra with a N/S value of 0.89

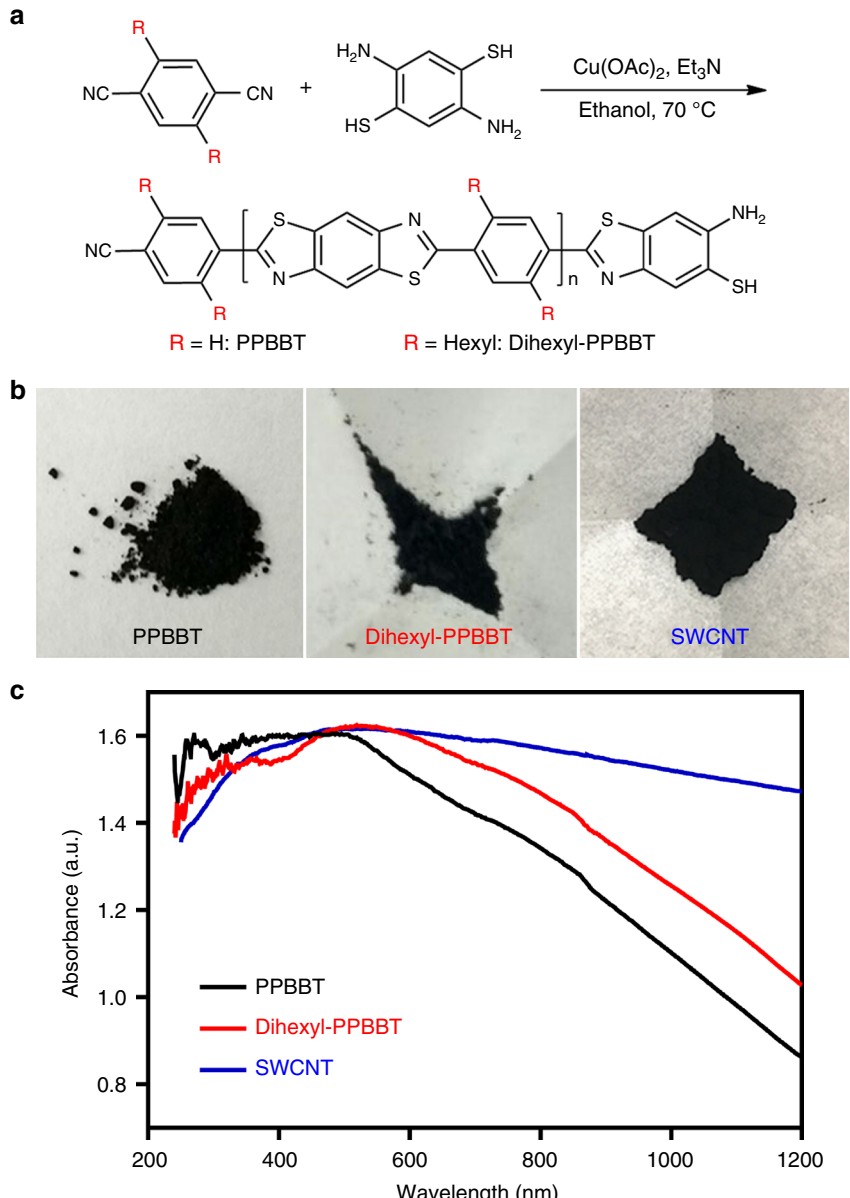

**Fig. 1** Syntheses and optical properties of PPBBT (or Dihexyl-PPBBT). **a** Synthetic routes for PPBBT and its dihexyl derivative Dihexyl-PPBBT. **b** Photographs of PPBBT, Dihexyl-PPBBT, and SWCNT powders. **c** UV-Vis-NIR absorption spectra of PPBBT, Dihexyl-PPBBT, and SWCNT

(Supplementary Figures 9 and 10, and Supplementary Table 3). These results demonstrated the feasibility and versatility of our approach in the facile syntheses of conjugated polymer materials.

To determine the optical properties of above-obtained conjugated polymers, UV-Vis-NIR spectra of PPBBT and Dihexyl-PPBBT were acquired, which consistently showed a broad absorption band from 250 to 1200 nm (Fig. 1c). Interestingly, their UV-Vis-NIR spectra are in good overlap with that of the solar spectrum[17] (Supplementary Figure 11), suggesting these two conjugated polymers could efficiently absorb and utilize solar energy. The energy band gap is a key parameter that controls the light absorption efficiency of the photothermal materials in solar cells[29]. Based on their corresponding UV-Vis-NIR spectra and using the equation $E = h\nu = 1240/\lambda$, the energy band gaps of our PPBBT and Dihexyl-PPBBT were roughly calculated to be 0.69 eV and 0.62 eV, respectively, which were near to that 0.54 eV of SWCNT. The low energy band gaps of PPBBT and Dihexyl-PPBBT suggest their potential application in

organic photovoltaics. After their light absorption properties were characterized, the electronic structures of PPBBT and Dihexyl-PPBBT were also characterized using ultraviolet photoelectron spectroscopy (UPS) (Supplementary Figures 12 and 13). UPS valence-band spectra together with secondary electron cutoffs revealed that the ionization potential for PPBBT and Dihexyl-PPBBT were −5.47 eV and −5.95 eV, respectively, suggesting their good oxidative stability.

**Photothermal properties of PPBBT.** Above characterizations indicated good photothermal properties of our conjugated polymers PPBBT and Dihexyl-PPBBT. To demonstrate their photothermal properties, we used a Xenon (Xe) lamp, whose full light spectrum closes to that of the Sun, to shine PPBBT using SWCNT as the control. As shown in Fig. 2a, when exposed to 0.8 W cm$^{-2}$ Xe light, PPBBT exhibited a comparable heating-up speed to that of SWCNT. The maximum photothermal temperature 71.0 °C of

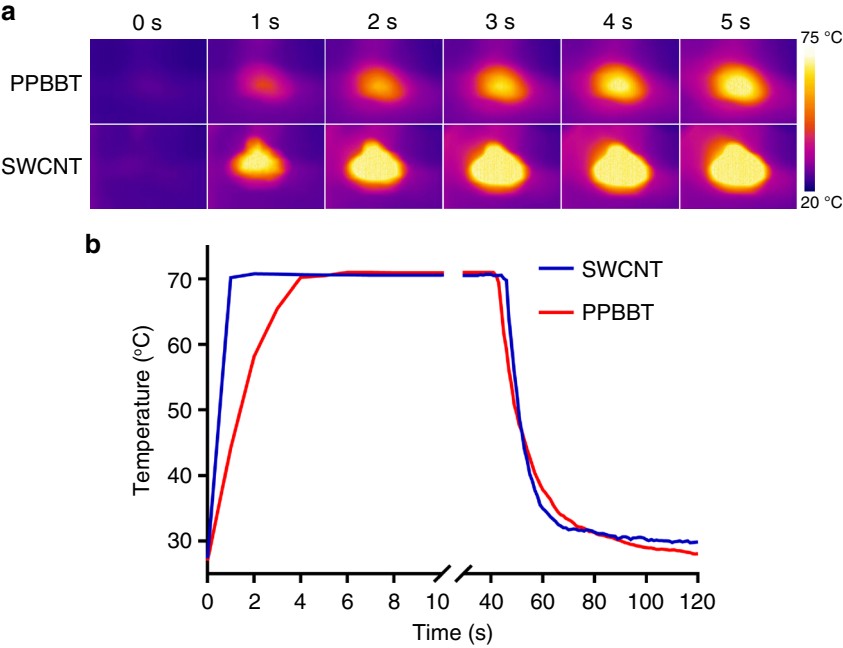

**Fig. 2** Photothermal properties of PPBBT. **a** Time-dependent thermal images of 100 mg PPBBT and 100 mg SWCNT powders exposed to 0.8 W cm$^{-2}$ Xe light. **b** Temperature curve as a function of Xe lamp irradiation time in a. The light source was removed at about 40 s

PPBBT was close to that 70.8 °C of SWCNT (Fig. 2b), implying PPBBT a good material for solar energy utilization. To calculate the photothermal conversion efficiency, we also measured the rate of temperature decrease after removing the light source at 40 s (Fig. 2b). The photothermal conversion efficiency of PPBBT was roughly calculated to be 94.3% of that of SWCNT[30], suggesting very good photothermal conversion property of our conjuagted polymer. Moreover, to further explore its potential photothermal applications in biomedical systems, we put PPBBT in a tube and used a thermal imaging system to record its temperature changes under NIR laser irradiation at 808 nm, which has deep tissue penetration ability and is frequently used in biomedicine[31]. SWCNT, star photothermal biomedical material known for its excellent thermal conductivity[32], was also employed for parallel study. As shown in Supplementary Figures 14 and 15, under continuous laser irradiation at 1.5 W cm$^{-2}$ and 808 nm, both PPBBT and SWCNT showed a quick temperature raise and reached their temperature plateaus at 160 s, suggesting their fast heating-up capability. The maximum photothermal temperature that PPBBT could reach under this condition was 107 °C, lower than that 160 °C of SWCNT. The photothermal conversion efficiency of PPBBT at 808 nm was roughly calculated to be 62.3% of that of SWCNT. We proposed that the better NIR performance of SWCNT is due to its higher absorption coefficient at NIR region than PPBBT. Nevertheless, we envision that PPBBT would still be a good candidate for biomedical applications considering its SWCNT-comparably high photothermal capability.

**In vitro photothermal properties of NP$_{PPBBT}$.** To investigate the potential PTT application of our photothermal materials for cancer treatment, we first managed to wrap PPBBT into water-soluble polymer nanoparticles. mPEG-*b*-PHEP, an amphiphilic diblock copolymer which has been successfully applied in many drug delivery systems, was used to fabricate the PPBBT-loaded nanoparticles (NP$_{PPBBT}$) through nanoprecipitation method (Fig. 3a)[33]. A dark yellow NP$_{PPBBT}$ monodispersion in water was obtained, which remained stable for at least 24 h (inset of Fig. 3b, Supplementary Figures 16 and 17). UV-Vis spectra showed that

43% of PPBBT was loaded into the nanoparticles (Supplementary Figure 18), suggesting an acceptable loading efficiency. The extinction coefficient of NP$_{PPBBT}$ in phosphate buffer saline (PBS) at 808 nm was calculated to be 7.03 L g$^{-1}$ cm$^{-1}$ (Supplementary Figure 19), slightly lower than that 20.2 L g$^{-1}$ cm$^{-1}$ of poly-pyrrole nanosheet recently reported[30]. Dynamic light scattering (DLS) showed the NP$_{PPBBT}$ had an average diameter of 107.0 ± 2.6 nm, while transmission electron microscopy (TEM) revealed the nanoparticles had a spherical morphology (Fig. 3b). To evaluate the photothermal properties of our black material after encapsulation, 50 μg mL$^{-1}$ NP$_{PPBBT}$ monodispersion in water was exposed to the NIR laser irradiation at 808 nm and a power density of 0.5, 0.75, 1.0, or 1.5 W cm$^{-2}$ for 10 min, and its dynamic temperatures were monitored by an infrared camera. As shown in Fig. 3c and Supplementary Figure 20, the temperatures of NP$_{PPBBT}$ solution rose rapidly, reached their plateaus at about 5 min, and the ΔT increased monotonically with the increase of laser power density (ΔT = 5.7 °C for 0.5 W cm$^{-2}$, 12.6 °C for 0.75 W cm$^{-2}$, 17.9 °C for 1.0 W cm$^{-2}$, and 24.9 °C for 1.5 W cm$^{-2}$ at 5 min). We then further investigated the photostability of NP$_{PPBBT}$. Three cycles of on/off NIR laser irradiation on the NP$_{PPBBT}$ tube were performed (808 nm, 1.0 W cm$^{-2}$, laser on for 5 min and remained off till room temperature). As shown in Fig. 3d, a temperature change of 16.6 °C of the tube was achieved after the first cycle of laser irradiation but did not show observable decrease after two additional cycles, indicating good photostability of NP$_{PPBBT}$. Then, as shown in Fig. 3e, heat transfer time constant (τ$_s$) of NP$_{PPBBT}$ was determined to be 140.63 s by linear correlation of the cooling times versus negative natural logarithm of driving force temperatures in Fig. 3d. Using obtained τ$_s$, the photothermal conversion efficiency (η)[30] of NP$_{PPBBT}$ was calculated to be 32.4% (Fig. 3e), which was comparable to those of reported PTT materials[34–36].

**Cell uptake and photo cytotoxicity of NP$_{PPBBT}$ nanoparticles.** With their good in vitro photothermal property verified, NP$_{PPBBT}$ nanoparticles were subsequently applied for in vivo PTT of tumour. Before that, 1,1′-dioctadecyl-3,3,3′,3′-tetramethylindodicarbocyanine,

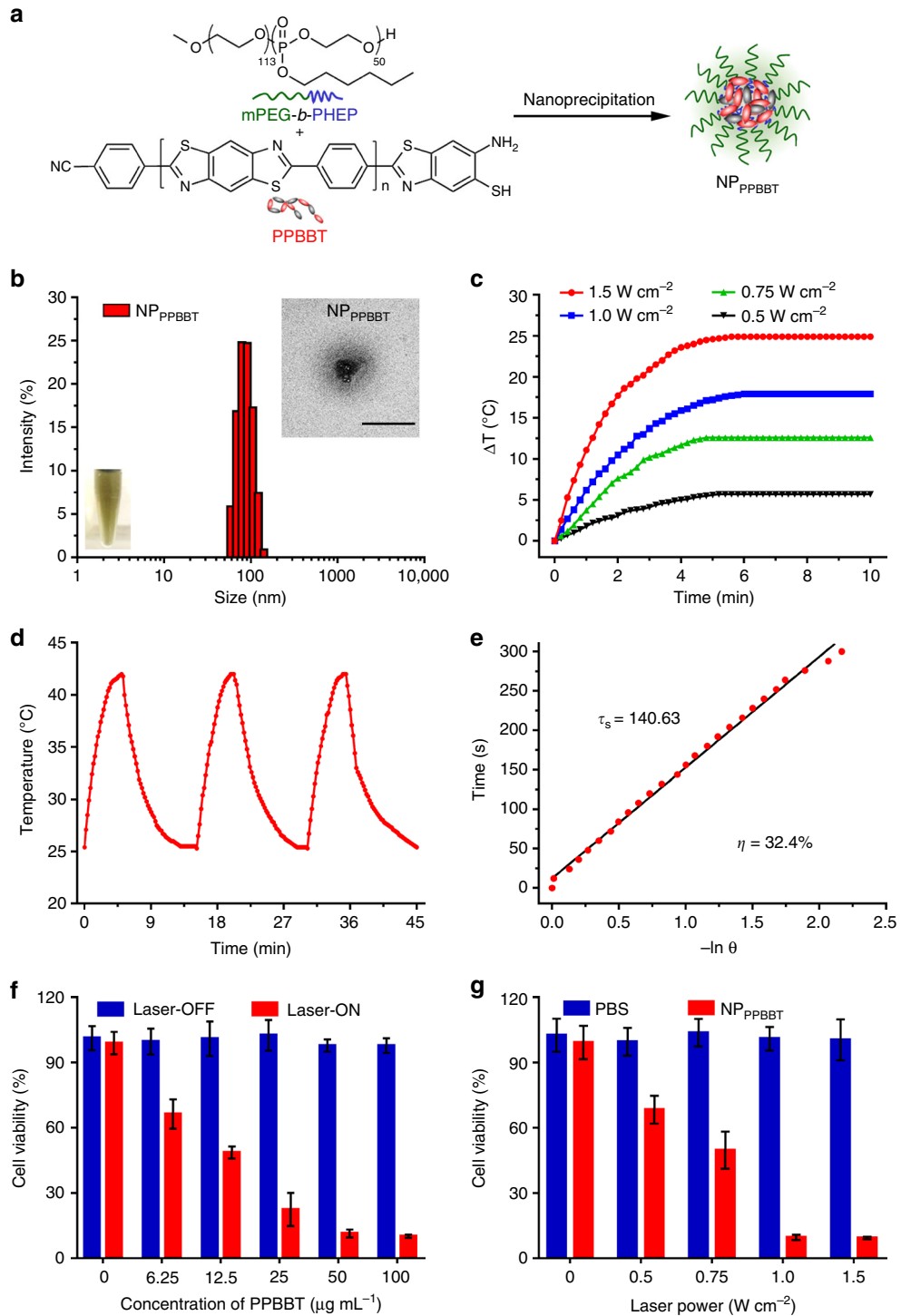

**Fig. 3** In vitro photothermal properties and photo cytotoxicity of NP$_{PPBBT}$. **a** Schematic illustration of the preparation of NP$_{PPBBT}$. **b** DLS measurement of NP$_{PPBBT}$. Insert: optical image (bottom left) and TEM image (top right) of NP$_{PPBBT}$. Scale bar, 200 nm. **c** Time-dependent temperature change curves of NP$_{PPBBT}$ solution upon exposure to the 808 nm NIR laser at a power density of 0.5, 0.75, 1.0, or 1.5 W cm$^{-2}$. **d** Temperature elevation curves of NP$_{PPBBT}$ over three cycles of 808 nm NIR laser on/off irradiation. **e** Linear correlation of the cooling times versus negative natural logarithm of driving force temperatures. **f** Relative photo (or dark) viabilities of EMT-6 cells after treatment with NP$_{PPBBT}$ at different concentrations with (or without) 808 nm NIR laser irradiation at 1.0 W cm$^{-2}$ for 10 min. **g** Relative photo viabilities of EMT-6 cells incubated with 50 μg mL$^{-1}$ NP$_{PPBBT}$ (or PBS) under 808 nm NIR laser irradiation at different power densities. Results are presented as mean ± S.D., $n = 3$

4-chlorobenzenesulfonate salt (DiD), a NIR fluorescent dye, was co-encapsulated with PPBBT to form NP$_{PPBBT}$/DiD nanoparticles to study the cell uptake property of NP$_{PPBBT}$ nanoparticles by EMT-6 breast cancer cells. NP$_{PPBBT}$/DiD nanoparticles were prepared using the same nanoprecipitation method as that of NP$_{PPBBT}$ nanopartciles

and the loading efficiency of DiD was calculated to be 83% (Supplementary Figure 21). Confocal laser scanning microscopy fluorescence images of the cells incubated with NP$_{PPBBT}$/DiD for 0.25, 0.5, 1, 2, 4, or 8 h showed that the red DiD fluorescence gradually increased inside cells and approached its plateau at about 4 h[37]

(Supplementary Figure 22). Whereafter, photo cytotoxicity of $NP_{PPBBT}$ nanoparticles on EMT-6 cells after 4 h incubation with the nanoparticles was evaluated using 3-(4,5-dimethylthiazol-2-yl) 2,5 diphenyltetrazplium bromide (MTT) assay. As shown in Fig. 3f, without NIR light illumination, $NP_{PPBBT}$ nanoparticles did not show observable cytotoxicity to the cells at the concentration up to 100 μg mL$^{-1}$. However, upon 10 min irradiation of 808 nm NIR laser at 1.0 W cm$^{-2}$, viability of the cells decreased with the increase of $NP_{PPBBT}$ concentration (from 0–100 μg mL$^{-1}$). Specifically, more than 88.6% cells were killed at $NP_{PPBBT}$ concentration of 50 μg mL$^{-1}$ under 1.0 W cm$^{-2}$ NIR laser irradiation (Fig. 3f). The photothermal cytotoxicity of the cells incubated with 50 μg mL$^{-1}$ $NP_{PPBBT}$ under different laser powers was also evaluated by MTT assay. After pre-incubation with 50.0 μg mL$^{-1}$ $NP_{PPBBT}$ for 4 h, the cells were exposed to NIR lasers at different power densities. As shown in Fig. 3g, as expected, cell viability decreased with the increase of laser power density. However, in the absence of $NP_{PPBBT}$, no obvious photo cytotoxicity was observed even at the highest power density of 1.5 W cm$^{-2}$.

**In vivo performance of $NP_{PPBBT}$ nanoparticles on tumours.** We then investigated the pharmacokinetics of $NP_{PPBBT}$ nanoparticles in healthy mice and orthotopic EMT-6 tumour-bearing mice.

$NP_{PPBBT}$/DiD was intravenously (i.v.) injected into the healthy mice or tumour-bearing mice at a DiD concentration of 0.25 mg kg$^{-1}$. Blood drug concentration-time curves of $NP_{PPBBT}$/DiD indicated that the elimination half life ($T_{1/2z}$) of $NP_{PPBBT}$ was about 95.4 h in healthy mice and 46.6 h in tumour-bearing mice (Fig. 4a, Supplementary Figure 23 and Supplementary Table 4) according to the noncomparement analysis (DAS 3.2.6). The long circulation property of $NP_{PPBBT}$/DiD might lead to its efficient accumulation in tumours due to its EPR effect[38], evidenced by that its $T_{1/2z}$ of 95.4 h in healthy mice was shortend to 46.6 h in tumour-bearing mice. To investigate the biodistribution of $NP_{PPBBT}$, the tumour-bearing mice were sacrificed at 12, 24, 48, or 72 h post i.v. injection of $NP_{PPBBT}$/DiD and the major organs (heart, liver, spleen, lung, kidney, and tumour) were imaged using an Xenogen IVIS® spectrum system. Both the biodistribution data in Fig. 4b, and the ex vivo fluorescence images and analyses in Supplementary Figures 24 and 25 of the major organs consistently indicated that, tumours had the highest average DiD fluorescence among all the major organs studied and their fluorescence reached to its maximum at 24 h then gradually decreased. Above results indicated that, due to tumour EPR effect, our $NP_{PPBBT}$ nanoparticles were effectively accumulated in tumours, which was favorable for their PTT on tumours. Accordingly, orthotopic EMT-6 tumour-bearing BALB/c mice were

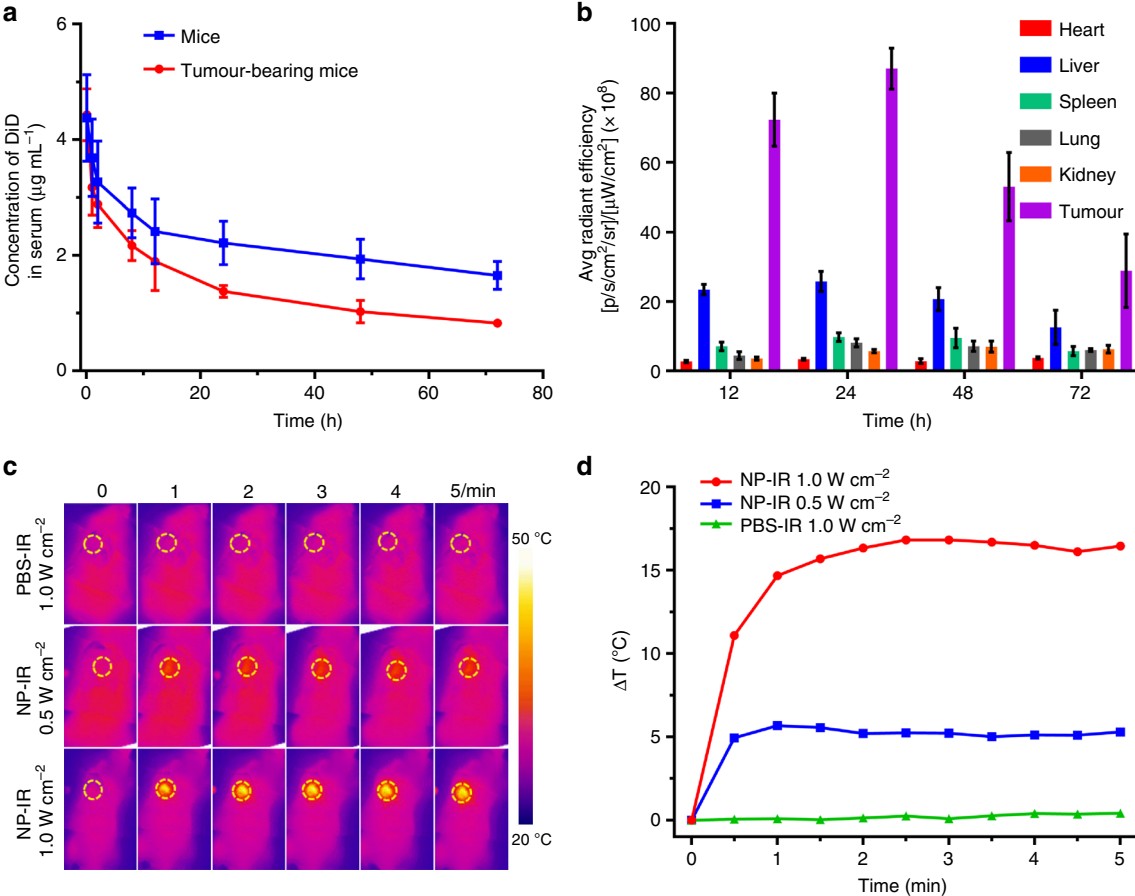

**Fig. 4** In vivo performance of $NP_{PPBBT}$ nanoparticles on tumours. **a** Blood DiD concentration *vs*. time curves in healthy mice or orthotopic EMT-6 breast tumour-bearing mice intravenously injected with $NP_{PPBBT}$/DiD at a DiD dose of 0.25 mg kg$^{-1}$ (*n* = 3 for each group). **b** Quantification of DiD fluorescence from the major organs (heart, liver, spleen, lung, kidney, and tumours) in orthotopic EMT-6 tumour-bearing mice sacrificed at 12, 24, 48 or 72 h post injection of $NP_{PPBBT}$/DiD at a DiD concentration of 0.25 mg kg$^{-1}$. **c, d** IR thermal images (the tumours were indicated by yellow dashed circles) (**c**) and tumour temperature evolutions (**d**) of orthotopic EMT-6 tumour-bearing mice after intravenous injection of PBS or 5 mg kg$^{-1}$ $NP_{PPBBT}$ under 808 nm laser irradiation at 0.5 or 1.0 W cm$^{-2}$ for 5 min. NP-IR: $NP_{PPBBT}$ plus laser irradiation; PBS-IR: PBS plus laser irradiation. Results are presented as mean ± S.D., *n* = 3

i.v. injected with $NP_{PPBBT}$ nanopartiles at a dose of 5 mg kg$^{-1}$. Twenty-four-hours post injection, the tumours in mice were irradiated with an 808 nm laser for 5 min at a power density of 0.5 or 1.0 W cm$^{-2}$. The spatial temperature distribution and the temperature increase at the tumour sites were real-time monitored with an IR imaging camera (Fig. 4c). As shown in Fig. 4d, the temperature of the tumour region of the $NP_{PPBBT}$-injected group raised quickly by 5.2 °C within 1 min under 0.5 W cm$^{-2}$ laser irradiation and 16.5 °C within 2 min under 1.0 W cm$^{-2}$ laser irradiation, respectively. However, the temperature of the tumour region of the PBS-injected control group did not show obvious increase even under 1 W cm$^{-2}$ laser irradiation (Fig. 4c, d).

**PTT efficiency of $NP_{PPBBT}$ nanoparticles on tumours.** Encouraged by the efficient temperature increase in tumour site, the PTT efficiency of $NP_{PPBBT}$ nanopartiles for anti-tumour treatment was further studied. Mice bearing orthotopic EMT-6 tumour were randomly divided into five groups (n = 5 per group) among which three groups of mice were intravenously (i.v.) injected with 5 mg kg$^{-1}$ $NP_{PPBBT}$ while two groups of mice were i.v. injected with PBS. Twenty-four-hours post injection, tumours of two experimental groups of $NP_{PPBBT}$-injected mice were irradiated with 808 nm laser for 10 min at a power density of 0.5 or 1.0 W cm$^{-2}$, respectively. Tumours of one group of $NP_{PPBBT}$-injected mice without laser irradiation and tumours of two groups of PBS-injected group mice with (or w/o) 1.0 W cm$^{-2}$ laser irradiation were designated as three control groups. During the treatment, body weights of the mice were monitored and the results indicated that none of above five-type treatments induced toxicity to the mice (Fig. 5a). Time course tumour volume curves indicated that the tumour growth in the experimental group of 5 mg kg$^{-1}$ $NP_{PPBBT}$ plus 1.0 W cm$^{-2}$ NIR laser irradiation was the most significantly inhibited (Fig. 5b). Lower laser density (0.5 W cm$^{-2}$) together with 5 mg kg$^{-1}$ $NP_{PPBBT}$ in the experimental group also showed significant inhibition on tumour growth but not as efficient as that of mice treated by higher laser density (1.0 W cm$^{-2}$). However, none of the three control groups (i.e. 5 mg kg$^{-1}$ $NP_{PPBBT}$ without laser irradiation, PBS with 1.0 W cm$^2$ laser irradiation, and PBS without laser irradiation) delayed the tumour growth (Fig. 5b), suggesting nethier the laser alone nor $NP_{PPBBT}$ nanopartiles alone could inhibit tumour growth. After monitoring the body weights and tumour volumes of the mice for 15 days, we sacrificed the mice, took photographs of the tumours (Fig. 5c), and weighted the tumours (Fig. 5d). The results indicated that the tumour growth in the experimental group of 5 mg kg$^{-1}$ $NP_{PPBBT}$ plus 1.0 W cm$^{-2}$ laser irradiation was the mostly inhibited (Fig. 5c, d) while that among three control groups was not significant (Fig. 5c, d). Above results indicated that our $NP_{PPBBT}$ had high PTT efficiency for anti-tumour treatment. Furthermore, hematoxylin-eosin (H&E) staining, Ki67 immunofluorescence staining[39], and terminal transferased UTP nick-end labeling (TUNEL) assay of the tumour tissues to analyze cell state, cell proliferation, and cell apoptosis in the tissues, respectively, were conducted after PTT treatment (Fig. 5e). As expected, the most significant tumour cell damage, the mostly reduced percentage of proliferating tumour cells, and the mostly increased percentage of apoptotic tumour cells were observed in the experimental group of 5 mg kg$^{-1}$ $NP_{PPBBT}$ plus 1.0 W cm$^{-2}$ laser irradiation (Fig. 5e). Moreover, H&E staining of the major organ tissues of the mice sacrificed at day 15 post treatment indicated that there was no obvious damage in all major organs, suggesting that $NP_{PPBBT}$ nanoparticles were biocompatible (Supplementary Figure 26). Meanwhile, the pharmacokinetics, biodistribution, and photothermal performance on tumours of $NP_{PPBBT}$ nanoparticles in subcutaneously EMT-6 tumour-bearing mice, as well as anti-tumour PTT efficiency of the nanoparticles on the tumour-bearing mice, were evaluated (Supplementary Figures 27–29). The results indicated that our $NP_{PPBBT}$ nanopartiles also exhibted excellent anti-tumour efficiency on subcutaneous breast tumour as well. All these results suggested that our $NP_{PPBBT}$ nanoparticles are suitable for PTT of tumours in vivo, pathologically and morphologically.

## Discussion

In summary, by modification of a CBT-Cys click condensation reaction and rational design of the starting materials, we facilely synthesized conjugated polymers with good photothermal properties and applied them for effective PTT of tumour. UV-Vis-NIR spectra of as-obtained PPBBT and Dihexyl-PPBBT exhibited good overlap with solar spectrum, suggesting their potential applications in solar energy ultilization. UV-Vis-NIR data analysis together with their UPS spectra further indicated that PPBBT and Dihexyl-PPBBT had low energy band gaps and high oxidative stability, respectively. Using Xe light to mimic sunlight, we verified that our PPBBT had comparable photothermal heating-up speed to that of SWCNT. Moreover, using a nanoprecipitation method, the hydrophobic PPBBT, together with diblock copolymer mPEG-*b*-PHEP, was employed to fabricate the water-soluble, biocompatible $NP_{PPBBT}$ nanoparticles, which maintained good photothermal conversion efficiency and photostability. In vivo tumour PTT experiments indicated, while neither 5 mg kg$^{-1}$ $NP_{PPBBT}$ nor 1.0 W cm$^{-2}$ laser irradiation alone could inhibit tumour growth, 5 mg kg$^{-1}$ $NP_{PPBBT}$ plus 1.0 W cm$^{-2}$ laser irradiation exhibited excellent PTT effect on the tumours. Compared with other conjugated polymers that commonly used for preparing PTT agents (e.g. polyaniline (PANI), poly-(3,4-ethylenedioxythiophene):poly(4-styrenesulfonate)    (PEDOT:PSS),    and polypyrrole (PPy))[40], the synthesis of our PPBB T is relatively faster and more facile. Meanwhile, the preparation procedure of the PPBBT-loaded water-soluble nanoparticles $NP_{PPBBT}$ is routine and PTT efficiency of our $NP_{PPBBT}$ nanoparticles on tumours is comparable to that of other modified conjugated polymers. We envision that this work would provide a facile approach to fabricate conjugated polymers for more promising applications in biomedicine or photovoltaics in the near future.

## Methods
**General methods.** All the starting materials were obtained from J&K Chemical Company (Shanghai). Commercially available reagents were used without further purification, unless noted otherwise. All chemicals were reagent grade or better. Commercial single-wall carbon nanotube was obtained from Shenzhen Nanotech Port Co. Fetal bovine serum (FBS) and Dulbecco's modified eagle medium (DMEM) were purchased from Gibco BRL (Eggenstein, Germany). 1,1′-diocta-decyl-3,3,3′,3′-tetramethylindodicarbocyanine, 4-chlorobenzenesulfonate salt (DiD) was purchased from Thermo Fisher Scientific (D7757). 4,6-diamidino-2-phenylindole (DAPI) was obtained from Sigma–Aldrich (D8417). Ultrapure water was generated from a Milli-Q Synthesis System (Millipore, Bedford, MA, USA). $^1$H NMR and $^{13}$C NMR spectra of 2,5-dihexyl-1,4-dicyanobenzene were obtained on a Bruker AV-300 MHz spectrometer. $^1$H NMR spectrum of PPBBT was obtained on a Bruker AV-400 MHz spectrometer. Molecular weight and molecular weight distribution of the polymer PPBBT with low solubility at room temperature were determined by gel permeation chromatography (GPC) with a PL 210 equipped with one Shodex AT-803S and two Shodex AT-806MS columns at 150 °C using trichlorobenzene as the eluent and calibrated with polystyrene standards. The electron ionization-mass spectrometry (EI-MS) spectrum of 2,5-dihexyl-1,4-dicyanobenzene was obtained on a Thermo Scientific™ Q Exactive™ GC Orbitrap™ GC-MS/MS equipped with a standard EI (70 eV) source. XPS experiments were performed on an ESCALAB 250 (Thermo-VG Scientific). UV-Vis-NIR absorption spectra were recorded on a DUV-3700 spectrophotometer (Shimadzu). Ultraviolet photoelectron spectroscopy (UPS) experiments were performed at the Catalysis and Surface Science End-station in National Synchrotron Radiation Laboratory (NSRL), Hefei. The valence-band spectra were measured using synchrotron-radiation light as the excitation source. The Xenon lamp (PL-X3000) was bought from Pu Lin Sai Si Company (Beijing). Transmission electron microscopy (TEM) was conducted on a JEM-ARM 200F Atomic Resolution Analytical Microscope operating at an accelerating voltage of 200 kV. Dynamic light scattering (DLS)

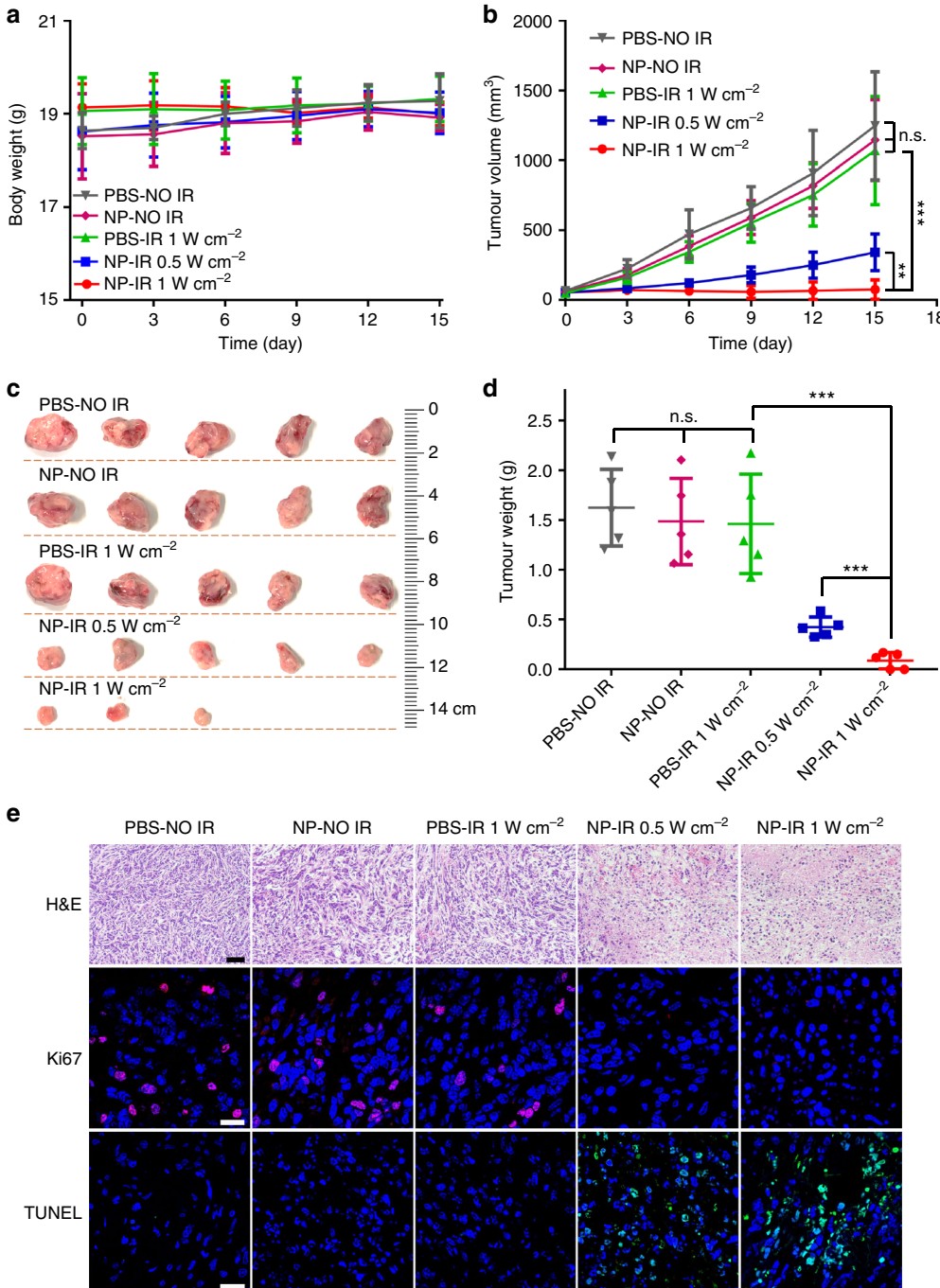

**Fig. 5** PTT efficiency of NP$_{PPBBT}$ nanopartiles on tumours. **a**, **b** Body weight (**a**) and tumour volume (**b**) curves of orthotopic EMT-6 tumour-bearing mice at different time points after receiving one dose of treatment with indicated formulations. At 24 h post injection, tumours were irradiated with (or w/o) laser for 10 min and the observations started. **c**, **d** Photographs (**c**) and weights (**d**) of tumours in mice sacrificed after 15-day observation. **e** H&E staining, Ki67 (Cy3, red) and DAPI (blue) counterstaining, and TUNEL (FITC, green) and DAPI (blue) counterstaining of the tumour tissues from mice sacrificed at day 15 post treatment. Scale bars, 50 μm for H&E staining; 20 μm for Ki67 and TUNEL staining. PBS-IR: PBS plus laser irradiation; NP-IR: NP$_{PPBBT}$ plus laser irradiation; NP-NO IR: NP$_{PPBBT}$ without laser irradiation. Results are presented as mean ± S.D., $n = 5$; n.s. not significant, $p > 0.05$; *$p < 0.05$; **$p < 0.01$; ***$p < 0.001$, analyzed by Student's $t$-test

spectra were recorded on a NanoBrook 90PLUS PALS particle size analyzer. Element contents were measured by an element analyzer (VarioELIII).

**Ultraviolet photoemission spectroscopy experiments**. The ultraviolet photoemission spectroscopy (UPS) examinations on the conjugated polymers were conducted at the photoemission end-station of BL10B beamline of the National Synchrotron Radiation Laboratory (NSRL) in Hefei, China. The valence-band spectra of the conjugated polymers were obtained when the materials were excited by the synchrotron-radiation light a photon energy of 169.02 eV. Photon energy of the beam

light was determined by referring to the Fermi level ($E_F = 0$) of Au. To observe the secondary electron cutoffs of the conjugated polymers, a sample bias of $-10$ V was used. The difference between the width of whole valence-band spectrum and the photon energy of each material was used to determine its work function ($\Phi$).

**Synthesis of 2,5-dihexyl-1,4-dicyanobenzene**. A solution of 1,4-dibromo-2,5-diethylbenzene (87 mg, 0.3 mmol) and CuCN (81 mg, 0.9 mmol) was refluxed in N, N'-dimethylformamide (DMF, 2 mL) for 24 h under nitrogen. The reaction mixture was poured into a saturated solution of NH$_4$OH yielding a brown precipitate.

The solid was filtered off, washed with $NH_4OH$ and water. The crude product was then purified with HPLC to obtain pure 2,5-dihexyl-1,4-dicyanobenzene (Supplementary Table 5). $^1H$ NMR (300 MHz, $CDCl_3$): δ 7.53 (s, 2 H), 2.87–2.74 (m, 4 H), 1.69–1.61 (m, 4 H), 1.41–1.25 (m, 12 H), 0.88 (t, $J = 6.9$ Hz, 6 H) (Supplementary Figure 6). $^{13}C$ NMR (75 MHz, $CDCl_3$): 143.8, 132.3, 115.7, 115.5, 32.8, 30.4, 29.9, 28.7, 21.5, 13.0 (Supplementary Figure 7). MS: calculated for $C_{20}H_{28}N_2$ $[M^+]$ = 296.22525, obsvd. EI-MS (70 eV): m/z 296.22453 (Supplementary Figure 8).

**Synthesis of diblock copolymer mPEG-*b*-PHEP**. The diblock copolymer mPEG-*b*-PHEP was synthetized according to the literature[37]. Briefly, a round bottom flask was freshly flamed and purged with nitrogen. PEG (0.500 g, 0.10 mmol, Mn = 5000 g mol$^{-1}$) and 2-hexoxy-2-oxo-1,3,2-dioxaphospholane (HEP, 1.42 g, 6.80 mmol) were added into above flask and dissolved in anhydrous tetrahydrofuran (9.1 mL). After the mixture was stirred for 10 min, 1,5,7-Triazabicylo [4.4.0]dec-5-ene (TBD, 13.9 mg, 0.10 mmol) was added. Then the mixture was stirred for additional 5 min and terminated with the addition of benzoic acid (0.122 g, 1.0 mmol). The clear reaction solution was concentrated, precipitated in a cold diethyl ether/methanol mixture (10/1, v/v), and filtered. The white crude product was purified by dissolving in a small volume of tetrahydrofuran and precipitated in above cold diethyl ether/methanol mixture again.

**Preparation of NP$_{PPBBT}$ nanoparticles**. PPBBT was loaded into the nanoparticles composed of mPEG-*b*-PHEP by a nanoprecipitation method. Briefly, mPEG-*b*-PHEP (10.0 mg) and PPBBT (1.0 mg) were dissolved in 2.0 mL DMSO, and then the solution was dropwise added into 10 mL ultrapure water under stirring. After stirring for 2 h, the solution was dialyzed against ultrapure water (MWCO: 14,000 Da) for 24 h to remove DMSO. The unloaded PPBBT was removed by centrifugation at $1150 \times g$, and the amount of PPBBT loaded in the nanoparticles was quantified by UV-Vis spectroscopy.

**Infrared thermal imaging of NP$_{PPBBT}$ nanoparticles**. The NP$_{PPBBT}$ nanoparticles with 50 μg mL$^{-1}$ PPBBT in Eppendorf tubes were irradiated by a 808-nm diode laser at a power density of 0.5, 0.75, 1.0, and 1.5 W cm$^{-2}$ (New Industries Optoelectronics, Changchun, China) for 10 min. The real-time temperatures and infrared images were recorded using an infrared camera (ICI7320, Infrared Camera Inc., Beaumont, Texas, USA) and analyzed using IR Flash thermal imaging analysis software (Infrared Cameras Inc.).

**Photostability of NP$_{PPBBT}$ nanoparticles**. The NP$_{PPBBT}$ nanoparticles (50.0 μg mL$^{-1}$) were exposed to an NIR laser (808 nm, 1.0 W cm$^{-2}$, 5 min, laser on). Subsequently, the NIR laser was turned off for 10 min, and the solution was naturally cooled to room temperature (laser off). The laser on and laser off cycles were repeated three times, and the change in temperature was monitored as described above, and then calculated of the photothermal conversion efficiency.

**Cell culture**. The mouse breast cancer cell line EMT-6 was obtained from the American Type Culture Collection (ATCC). The cells were cultured in complete DMEM medium (containing 10% FBS) at 37 °C in a 5% $CO_2$ atmosphere.

**Cell uptake of nanoparticles**. At first, the NP$_{PPBBT}$/DiD was prepared. mPEG-*b*-PHEP (10.0 mg), DiD (0.025 mg) and PPBBT (1.0 mg) were dissolved in 2.0 mL DMSO, and then the solution was dropwise added into 10 mL ultrapure water under stirring. After stirring for 2 h, the solution was dialyzed against (MWCO: 14,000 Da) ultrapure water for 24 h to remove DMSO.The EMT-6 breast cancer cells were seeded on coverslips in 24-well plates overnight. The medium was replaced with fresh complete medium containing NP$_{PPBBT}$/DiD nanoparticles (50.0 μg mL$^{-1}$). After being incubated for 0.25, 0.5, 1, 2, 4, or 8 h at 37 °C, the cells were washed with PBS, fixed with 1% formaldehyde, and counterstained with Alexa Fluor$^{TM}$ 488 phalloidin (Invitrogen, Thermo Fisher, A12379, diluted 100 times) and 10 μg mL$^{-1}$ DAPI. Finally, the slices were observed by Zeiss LSM 880 confocal laser scanning microscope (Carl Zeiss, Oberkochen, Germany).

**In vitro photo cytotoxicity of NP$_{PPBBT}$ nanoparticles**. EMT-6 cells were seeded in a 96-well plate ($3 \times 10^3$ cells per well) at an atmosphere of 37 °C and 5% $CO_2$ for 12 h. Then the medium was replaced with a fresh NP$_{PPBBT}$ nanoparticle-containing medium at different PPBBT concentrations and the cells were incubated for another 4 h. The cells were washed with PBS for three times, added with fresh medium, and then exposed to the NIR laser (808 nm, 1.0 W cm$^{-2}$) for 10 min. After that, the cells were incubated for additional 48 h and their viability was analyzed using MTT assay. Photothermal cytotoxicity of NP$_{PPBBT}$ nanoparticles at different power densities was determined as following. EMT-6 cells were seeded in 96-well plates at the atmosphere described as above. After the cells were incubated overnight and washed with PBS for three times, fresh NP$_{PPBBT}$ nanoparticle-containing medium at 50.0 μg mL$^{-1}$ was added. After an additional 4 h incubation, the cells were subjected to laser irradiation at different power density for 10 min. After further 48 h incubation of the cells, their viability was analyzed using MTT assay.

**Statements for the animal experiments**. All the animals received tender care in compliance with the guidelines outlined in the Guide for the Care and Use of Laboratory Animals. The procedures (including the tumours of the mice were irradiated with 808 nm laser for 10 min at a power density of 0.5 or 1.0 W cm$^{-2}$) were approved by the University of Science and Technology of China Animal Care and Use Committee with an affidavit of Approval of Animal Ethical and Welfare number of USTCACUC1801001.

**Animal tumour model**. Female BALB/c mice (6–8 weeks old) were purchased from Vital River Laboratories (Beijing, China). EMT-6 cells ($2 \times 10^5$ for each mouse) were injected into the mammary fat pat of female BALB/c mice to establish an orthotopic EMT-6 tumour model or subcutaneous injected into the right thigh of each mice to establish a subcutaneous EMT-6 tumour model. The tumour-bearing mice used for subsequent experiments, until the tumour volume reached 50–80 mm$^3$. Tumour volume was calculated as (tumour length) × (tumour width)$^2$ × 0.5.

**In vivo pharmacokinetics studies and tissue biodistribution**. Mice bearing EMT-6 tumours were intravenously injected with NP$_{PPBBT}$/DiD as described above. At different time point, the blood was taken from mouse eyelids and the major organs from tumour-bearing mice were detected by Xenogen IVIS® spectrum system. Pharmacokinetic parameters were obtained by noncomperement analysis (DAS 3.2.6).

**Tumour temperature monitoring during laser irradiation**. Orthotopic EMT-6 tumour-bearing BALB/c mice were intravenously injection with NP$_{PPBBT}$ at 5 mg kg$^{-1}$. After 24 h, the tumours were irradiated by NIR laser-808 nm at power density of 0.5 and 1.0 W cm$^{-2}$ for 5 min. Finally, the real-time temperatures and infrared images were recorded using an infrared camera and analyzed using IR Flash thermal imaging analysis software.

**In vivo photothermal therapeutic efficacy of NP$_{PPBBT}$**. Orthotopic EMT-6 tumour-bearing mice were randomly divided into five groups ($n = 5$ per group) and were intravenously injected with NP$_{PPBBT}$ (5 mg kg$^{-1}$) or PBS. After 24 h of injection, the tumours were irradiated by an 808 nm laser at a power density of 0.5 and 1.0 W cm$^{-2}$ for 10 min. The tumour sizes were measured by a caliper every third day, and the mouse weights were recorded every third day.

**Immunohistochemical staining analysis**. At the end of photothermal therapy, the mice were sacrificed and tumour tissues were excised, fixed in 4% paraformaldehyde, and embedded in paraffin for analysis. Cell state in tumour tissue was analyzed by hematoxylin-eosin (H&E) staining. Cell proliferation and apoptosis in tumour tissue were also analyzed by immunofluorescence staining of the Ki67 antigen (Ki-67 (D2H10) Rabbit mAb (IHC Specific), Cell Signaling Technology, #9027, diluted 100 times; Goat Anti-Rabbit IgG (H + L) Secondary Antibody, Cy3 Conjugate, BOSTER, SKU: BA1032, diluted 100 times), and the terminal transferased UTP nick-end labeling (TUNEL) assay (In Situ Cell Death Detection Kit, Fluorescein, Roche, 11684795910, diluted 50 times). The major organs (heart, liver, spleen, lung, and kidney) were sectioned and analyzed by H&E staining for biocompatibility evaluation.

## Data availability

The data that support the findings of this study are available from the corresponding author upon reasonable request.

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

## Acknowledgements

The authors are grateful to the National Synchrotron Radiation Laboratory in the University of Science and Technology of China for UPS test. This work was supported by Ministry of Science and Technology of China (2016YFA0400904), the National Natural Science Foundation of China (Grants 21725505, 81821001, 21675145, 51773191 and 51573176).

## Author contributions

P.C., Y.M. and Z.Z. contributed equally to this work. They performed syntheses, characterizations, photothermal studies and photothermal therapy of mice. C.W. helped with the synthesis of the materials. Y.W. helped with project design and paper preparation. G.L. designed this project and wrote the paper.

## Additional information

**Competing interests:** The authors declare no competing interests.

