## [Peer Review File · Nature Communications]

Reviewers' comments:

Reviewer #1 (Remarks to the Author):

Title: Facile syntheses of conjugated polymers for photothermal therapy of tumor

Manuscript ID: NCOMMS-18-33848

Authors: Peiyao Chen, Yinchu Ma, Zhen Zheng, Chengfan Wu, Yucai Wang, and Gaolin Liang

Comments:

Chen et al. reported a novel conjugated polymer that could be synthesized via a facile method. The PPBBT based conjugated polymer had comparable photothermal properties with single-wall carbon nanotubes, which makes them capable to ablate cancer cells both in vitro and in vivo. Conjugated polymers have been broadly applied in the biomedical field owing to their high biocompatibility and potential biodegradability. Although a facile method was used to synthesize the conjugated polymer in this work, their advantages over carbon nanotubes or other photothermal agents were not clear. The overall novelty for this manuscript is not high, which might not be of interest to broad audiences in this field, so I recommend publishing in a more specialized journal. Here are some suggestions regarding the design and data presented.

1) To compare the heating-up speed between PPBBT and SWCNT, did they have the same mass concentration or the same absorbance at a specific wavelength in the NIR region? As the heating-up speed of photothermal agents is concentration dependent, it is hard to find a fair way to compare the completely different materials.

2) For the pharmacokinetic study, instead of using the percentage of injected dosage, it is better to use the absolute concentration of nanoparticles in the serum at different time points to reflect its real circulation half-life, which can be compared with other nanoparticles. In addition, tumor-bearing mice are not normally used for pharmacokinetic study as the EPR effect might shorten the half-life of the nanoparticles.

3) The biodistribution data in Figure 4B showed the tumor has the highest nanoparticle accumulation in 24 hours after injection. However, the mice were all treated 12 hours after injection in Figure 5B. Moreover, the saline injection only group should be added as a blank control, which aims to show if the laser alone (1 w/cm²) or nanoparticle alone would affect the tumor growth.

Reviewer #2 (Remarks to the Author):

In this manuscript, Peiyao, Yinchu, Zhen and coauthors applied the concept of the rapid condensation reaction between 2-cyano-benzothiazole (CBT) and the 1,2-aminothiol group of cysteine for design and facile synthesis of a new conjugated polymer poly-2-phenyl-benzobisthiazole (PPBBT). They characterized PPBBT in a series of studies in vitro, showed its comparable photothermal properties to existing conjugated polymers, and further generated PPBBT nanoparticles (NPPBBT) for photothermal treatment of a human breast cancer xenograft. Before this study, many conjugated polymers with good photothermal properties have been reported and utilized for photothermal therapy of cancer. Nevertheless, the rapid CBT-Cys condensation reaction which enables facile synthesis and provides great potential for further modification has never been utilized to design and synthesize photothermal materials. This study is interesting and overall well structured. However, the lack of some key data in terms of the polymer synthesis remains the major flaw as described in detail below. Together, I recommend a major revision for this manuscript.

Specific comments:

1. Reference 10 is not for photothermal therapy of cancer. May not be proper to cite here, but a minor issue.
2. NMR spectra of PPBBT and its analogs are required to confirm the chemical structures.
3. Gel Permeation Chromatography to measure the molecular weight or distribution of PPBBT and

its analogs are required.

4. Absorption coefficient of the polymers at the excitation wavelength, and comparison to existing polymers or carbon tubes need to be presented.

5. Mass data for Scheme S1 are required.

6. Compared to PCB1 [photothermal conversion efficiency: 42.8% (Ref 33)] and PBIBDF-BF nanoparticles [photothermal conversion efficiency: 46.7% (Ref 32)], the photothermal conversion efficiency of NPPPBBT reported here is 32.4%. The authors need to justify the advantage of using NPPPBBT for photothermal therapy of cancer instead of PCB1 and PBIBDF-BF.

7. For in vitro photo cytotoxicity study, the uptake of NPPPBBT by EMT6 cells need to be determined. From Method, the cell medium with particles was not replaced after 4 hours incubation, raising the question whether the particles were taken into cells or just on cell surface.

8. For ex vivo IHC staining of tumor slides in figure 5E: First of all, the resolution is relatively low; second, I understand H&E, part of the TUNEL staining, but not Ki67. For TUNEL, I think the blue may be DAPI and the green may be the double strand DNA breaks. It needs description. For Ki67, if the blue indicates nucleus and brown staining indicate Ki67 protein, why PBS-IR group showed significantly less nucleus but stronger proliferation staining?

Reviewer #3 (Remarks to the Author):

The manuscript "Facile syntheses of conjugated polymers for photothermal therapy of tumor" by Chen et al developed a new conjugated polymer with facile method and systematically explore its therapeutic efficacy. This study was well designed and there are convincing evidences to support the conclusions. Therefore, I recommend the acceptance of this manuscript after minor revision.

Major points:

1. In this study, authors used subcutaneous breast tumor model to systematically evaluate the therapeutic efficacy of the developed conjugated polymers. Actually, there are big differences between subcutaneous tumors and orthotopic ones. Therefore, I wonder whether authors could provide related data using orthotopic breast tumor model. This will greatly approve the unique advantages of the developed photothermal nanoplatform.

2. On page 13, line 14, it was mentioned that EMT-6 tumor-bearing nude mice were used in this study for the cancer treatment. I recommend authors to check whether it is nude mice or not.

3. Recently, accumulating evidences indicated that nanoplatform-based photothermal therapy could induce antitumor immune responses to combat tumor progression (Nat Commun, 2016, 7, DOI:10.1038/ncomms13193). In addition, it was evidenced that some conjugated polymers could serve as vaccine adjuvant to activate immune cells (Nanoscale, 2015, 7, 19282-19292). Therefore, I recommend authors to give more comprehensive discussions in the introduction.

Minor points:

1. On page 5, line 18, "...modified with additional functional groups..." should be "...modified with additional functional groups...".

2. On page 9, line 6, "To caculate the photothermal conversion efficiency, ..." should be "To calculate the photothermal conversion efficiency, ...".

The following are our responses to the comments (*in Italics*) of reviewers and the changes (underlined) made in this manuscript (NCOMMS-18-33848).

Reviewer #1

[Chen et al. reported a novel conjugated polymer that could be synthesized via a facile method. The PPBBT based conjugated polymer had comparable photothermal properties with single-wall carbon nanotubes, which makes them capable to ablate cancer cells both in vitro and in vivo. Conjugated polymers have been broadly applied in the biomedical field owing to their high biocompatibility and potential biodegradability. Although a facile method was used to synthesize the conjugated polymer in this work, their advantages over carbon nanotubes or other photothermal agents were not clear. The overall novelty for this manuscript is not high, which might not be of interest to broad audiences in this field, so I recommend publishing in a more specialized journal. Here are some suggestions regarding the design and data presented.]

We thank the reviewer for his/her assessment of this work. We agree with the reviewer that conjugated polymers have been broadly applied in the biomedical field owing to their high biocompatibility and potential biodegradability, which makes we believe that our work would be broadly interested.

As for novelty, **the rapid CBT-Cys click condensation reaction**, which was developed by me at Stanford University (*Nat. Chem.* 2010, 2, 54-60) and able to facilely synthesize new chemicals, **has never been utilized to design and synthesize conjugated polymers**. The benzobisthiazole and phenyl rings in the polymer backbone result in an extended delocalization of π electrons. Therefore, special electrical and optical properties derived from these materials are greatly anticipated, especially the electric and electrochromic properties. Moreover, **this conjugated polymer has never been used for photothermal therapy of tumors**. Considering abovementioned to novelties and after addressing the issues raised by the reviewer, we do think current revised manuscript could be considered for publication in *Nature Communications*.

[1. To compare the heating-up speed between PPBBT and SWCNT, did they have the same mass concentration or the same absorbance at a specific wavelength in the NIR region? As the heating-up speed of photothermal agents is concentration dependent, it is hard to find a fair way to compare the completely different materials.]

We thank the reviewer for him/her pointing out such important issues. Following the reviewer's suggestion, in current revision, we used the same mass (100 mg) of **PPBBT** and

SWCNT powders for the time-dependent photothermal study. As shown in current Figure 2, when exposed to 0.8 W/cm² Xe light, **PPBBT** exhibited a comparable heating-up speed to that of SWCNT. And the photothermal conversion efficiency of **PPBBT** was roughly calculated to be 94.3% of that of SWCNT according to the temperature decrease curves. These results suggested very good photothermal conversion property of our conjugated polymer. We revised the manuscript accordingly.

Original:

Page 9, line 4: The maximum photothermal temperature ~~71.9~~ °C of **PPBBT** was close to that ~~72.8~~ °C of SWCNT (Figure 2B), implying **PPBBT** a good new material for solar energy utilization. To calculate the photothermal conversion efficiency, we also measured the rate of temperature decrease after removing the light source at 40 s (Figure 2B). The photothermal conversion efficiency of **PPBBT** was roughly calculated to be ~~51.4~~ % of that of SWCNT⁴⁹.

Revision:

Page 9, line 13: The maximum photothermal temperature 71.0 °C of **PPBBT** was close to that 70.8 °C of SWCNT (Figure 2B), implying **PPBBT** a good new material for solar energy utilization. To calculate the photothermal conversion efficiency, we also measured the rate of temperature decrease after removing the light source at 40 s (Figure 2B). The photothermal conversion efficiency of **PPBBT** was roughly calculated to be 94.3 % of that of SWCNT³⁰, suggesting very good photothermal conversion property of our conjugated polymer.

Original Figure 2:

Revised Figure 2:

Figure 2 Photothermal properties of PPBBT. (A) Time-dependent thermal images of 100 mg PPBBT and 100 mg SWCNT powders exposed to 0.8 W/cm^2 Xe light. (B) Temperature curve as a function of Xe lamp irradiation time in A. The light source was removed at about 40 s.

[2. For the pharmacokinetic study, instead of using the percentage of injected dosage, it is better to use the absolute concentration of nanoparticles in the serum at different time points to reflect its real circulation half-life, which can be compared with other nanoparticles.]

We thank the reviewer for his/her instructive suggestion. Following the instructions from the reviewer, in current version, we calculated the absolute concentration of nanoparticles in the serum at different time points instead of using the percentage of injected dosage for the pharmacokinetic study of NP_{PPBBT} . Please comparatively check current Figure S25A with original Figure 4A.

Original Figure 4:

Current Figure S25:

[In addition, tumor-bearing mice are not normally used for pharmacokinetic study as the EPR effect might shorten the half-life of the nanoparticles.]

We thank the reviewer for his/her very instructive comments. Following his/her instruction, in current version, we further conducted pharmacokinetic study on healthy BALB/c mice and compared the results with those of the BALB/c mice bearing orthotopic EMT-6 tumors. As shown in current Figure 4A, those two blood drug concentration-time curves of NP_{PPBBT}/DiD indicated that the elimination half lives ($T_{1/2z}$) of NP_{PPBBT} were about 95.4 h in healthy BALB/c mice and 46.6 h in orthotopic EMT-6 tumor-bearing mice, respectively. We revised accordingly.

Original:

Page 12, line 4 from the bottom: We then investigated the pharmacokinetics and ~~biodistribution~~ of NP_{PPBBT} nanoparticles in EMT-6 ~~breast~~ tumor-bearing mice. ~~1,1'-dioctadecyl 3,3,3',3'-tetramethylindodicarbocyanine, 4-chlorobenzenesulfonate salt (DiD), a NIR fluorescent dye, was used to track the biodistribution of NP_{PPBBT} nanoparticles *in vivo*. NP_{PPBBT}/DiD was prepared using the same nanoprecipitation method as that of NP_{PPBBT} and intravenously (*i.v.*) injected into the tumor-bearing mice at a DiD concentration of 1.25 mg•kg⁻¹. Blood drug concentration-time ~~curve~~ of NP_{PPBBT}/DiD indicated that the elimination half life ($t_{1/2}$) of NP_{PPBBT} was about 10.32 h (Figure 4A), which was comparable to those of other reported PTT materials^{34,35}.~~

Revision:

Page 13, the last line: We then investigated the pharmacokinetics of NP_{PPBBT} nanoparticles in healthy mice and orthotopic EMT-6 tumor-bearing mice. NP_{PPBBT}/DiD was intravenously (*i.v.*) injected into the healthy mice or tumor-bearing mice at a DiD concentration of 0.25 mg•kg⁻¹. Blood drug concentration-time curves of NP_{PPBBT}/DiD indicated that the elimination half life ($T_{1/2z}$) of NP_{PPBBT} was about 95.4 h in healthy mice and 46.6 h in tumor-bearing mice (Figure 4A and Supplementary Table S4) according to the noncompartment analysis (DAS 3.2.6). The long circulation property of NP_{PPBBT}/DiD might lead to its efficient accumulation in tumors due to its EPR effect³⁸, evidenced by that its $T_{1/2z}$ of 95.4 h in healthy mice was shortend to 46.6 h in tumor-bearing mice.

38 Xu, X., Ho, W., Zhang, X., Bertrand, N. & Farokhzad, O. Cancer nanomedicine: from targeted delivery to combination therapy. *Trends Mol. Med.* **21**, 223-232 (2015).

Original Figure 4:

Revised Figure 4:

Figure 4 Pharmacokinetics, biodistribution, and photothermal performance in tumors of NP_{PPBTT} nanoparticles *in vivo*. (A) Blood DiD concentration vs. time curves in healthy mice or orthotopic EMT-6 breast tumor-bearing mice intravenously injected with NP_{PPBTT}/DiD at a DiD dose of $0.25 \text{ mg} \cdot \text{kg}^{-1}$ ($n = 3$ for each group). (B) Quantification of DiD fluorescence from the major organs (heart, liver, spleen, lung, kidney, and tumors) in orthotopic EMT-6 tumor-bearing mice

sacrificed at 12, 24, 48, or 72 h post injection of **NP_{PPBBT}/DiD** at a DiD concentration of 0.25 mg•kg⁻¹. IR thermal images (C) and tumor temperature evolutions (D) of orthotopic EMT-6 tumor-bearing mice after intravenous injection of PBS or 5 mg•kg⁻¹ **NP_{PPBBT}** under 808 nm laser irradiation at 0.5 or 1.0 W/cm² for 5 min.

Current Figure S22:

Figure S22. Fitted calibration curve of the average radiant efficiency of DiD dissolved in fetal bovine serum at different concentrations. This curve was used to calculate the serum DiD concentration in Figure 4A.

Current Table S4:

Table S4. Pharmacokinetic parameters of **NP_{PPBBT}/DiD** nanoparticles intravenously administered to the healthy mice or orthotopic EMT-6 tumor-bearing mice in Figure 4A. The data were obtained by noncompartment analysis (DAS 3.2.6).

Parameter	C _{max} (µg/mL)	T _{1/2z} (h)	AUC _{0-72 h} (µg/L•h)	CL _Z (L/h/kg)
Healthy mice	92.8	95.4	3131.6	0.716
Orthotopic EMT-6 tumor-bearing mice	88.6	46.6	2018.6	1.73

C_{max}, Peak concentration;
AUC, Area under the curve;
T_{1/2z}, Elimination half life;
CL_Z, Clearance rate.

[3. The biodistribution data in Figure 4B showed the tumor has the highest nanoparticle accumulation in 24 hours after injection. However, the mice were all treated 12 hours after injection in Figure 5B.]

We thank the reviewer for him/her pointing out such important issue. We are sorry that, in original Figure 4B, the biodistribution data was the **total** radiant efficiency of the major organs but we labeled them as the **average** radiant efficiency. Following his/her suggestion,

we reconducted the biodistribution study of NP_{PPBBT} in orthotopic breast tumor-bearing mice up to 72 h post injection. And in current revision, we used the **average radiant efficiency** data of the major organs to construct the Figure 4B. As consistently shown by the biodistribution data in current Figure 4B and the *ex vivo* fluorescence images of major organs in current Supplementary Figure S23, DiD fluorescence in tumors reached to its maximum at 24 h and then gradually decreased. Using this result, in current revision, we reconducted the PTT experiments on orthotopic EMT-6 tumor-bearing mice at 24 h post *i.v.* injection of NP_{PPBBT}. Please check current Figure 5. We revised the manuscript accordingly.

Original:

Page 13, line 5: To investigate the biodistribution of NP_{PPBBT}, the tumor-bearing mice were sacrificed at ~~6, 12, or 24 h~~ post *i.v.* injection of NP_{PPBBT}/DiD and the major organs (heart, liver, spleen, lung, kidney, and tumor) were imaged using an Xenogen IVIS® spectrum system. ~~As shown in Figure 4B and Supplementary Figure S16, DiD fluorescence imaging of the organs indicated that the NP_{PPBBT} nanoparticles were mainly located in the liver followed by the tumor. However, the fluorescence of DiD in tumor increased with time, indicating the effective accumulation of the NP_{PPBBT} nanoparticles in the tumor site due to the EPR effect, which was favorable for cancer treatment.~~ Accordingly, EMT-6 tumor-bearing BALB/c nude mice were *i.v.* injected with NP_{PPBBT} nanoparticles at a dose of 5 mg•kg⁻¹. ~~Twelve~~ hours post-injection, the tumors in mice were irradiated with an 808 nm laser for 5 min at a power density of 0.5 or 1.0 W/cm². The spatial temperature distribution and the temperature increase at the tumor sites were real-time monitored with an IR imaging camera (Figure 4C). As shown in Figure 4D, the temperature of the tumor region of the NP_{PPBBT}-injected group raised quickly by ~~7.5~~ °C within 3 min under 0.5 W/cm² laser irradiation and ~~16.7~~ °C within 2 min under 1.0 W/cm² laser irradiation, respectively.

Revision:

Page 14, line 9: To investigate the biodistribution of NP_{PPBBT}, the tumor-bearing mice were sacrificed at 12, 24, 48, or 72 h post *i.v.* injection of NP_{PPBBT}/DiD and the major organs (heart, liver, spleen, lung, kidney, and tumor) were imaged using an Xenogen IVIS® spectrum system. Both the biodistribution data in Figure 4B and the *ex vivo* fluorescence images in Supplementary Figure S23 of the major organs consistently indicated that, tumors had the highest average DiD fluorescence among all the major organs studied and their fluorescence reached to its maximum at 24 h then gradually decreased. Above results indicated that, due to tumor EPR effect, our NP_{PPBBT} nanoparticles were effectively accumulated in tumors which was favorable for their PTT on tumors. Accordingly, orthotopic EMT-6 tumor-bearing BALB/c mice were *i.v.* injected with NP_{PPBBT} nanoparticles at a dose of 5 mg•kg⁻¹. Twenty-four hours

post injection, the tumors in mice were irradiated with an 808 nm laser for 5 min at a power density of 0.5 or 1.0 W/cm². The spatial temperature distribution and the temperature increase at the tumor sites were real-time monitored with an IR imaging camera (Figure 4C). As shown in Figure 4D, the temperature of the tumor region of the **NP_{PPBBT}**-injected group raised quickly by 5.2 °C within 1 min under 0.5 W/cm² laser irradiation and 16.5 °C within 2 min under 1.0 W/cm² laser irradiation, respectively.

Original:

Page 14, line 10: ~~Twelve~~ hours post injection, tumors of two experimental groups of **NP_{PPBBT}**-injected mice were irradiated with 808 nm laser for 10 min at a power density of 0.5 or 1.0 W/cm², respectively.

Revision:

Page 15, line 5 from the bottom: Twenty-four hours post injection, tumors of two experimental groups of **NP_{PPBBT}**-injected mice were irradiated with 808 nm laser for 10 min at a power density of 0.5 or 1.0 W/cm², respectively.

Original:

Page S5, line 9 from the bottom: After ~~12~~ h, the tumors were irradiated by NIR laser-808 nm at power density of 0.5 and 1.0 W/cm² for 5 min.

Revision:

Page S6, line 12: After 24 h, the tumors were irradiated by NIR laser-808 nm at power density of 0.5 and 1.0 W/cm² for 5 min.

Original:

Page S5, line 2 from the bottom: After ~~12~~ h of injection, the tumors were irradiated by an 808 nm laser at a power density of 0.5 and 1.0 W/cm² for 10 min.

Revision:

Page S6, line 4 from the bottom: After 24 h of injection, the tumors were irradiated by an 808 nm laser at a power density of 0.5 and 1.0 W/cm² for 10 min.

Original Figure 4:

Revised Figure 4:

Figure 4 Pharmacokinetics, biodistribution, and photothermal performance in tumors of NP_{PPBTT} nanoparticles *in vivo*. (A) Blood DiD concentration vs. time curves in healthy mice or orthotopic EMT-6 breast tumor-bearing mice intravenously injected with NP_{PPBTT}/DiD at a DiD dose of $0.25 \text{ mg} \cdot \text{kg}^{-1}$ ($n = 3$ for each group). (B) Quantification of DiD fluorescence from the major organs (heart, liver, spleen, lung, kidney, and tumors) in orthotopic EMT-6 tumor-bearing mice

sacrificed at 12, 24, 48, or 72 h post injection of $\text{NP}_{\text{PPBBT}}/\text{DiD}$ at a DiD concentration of $0.25 \text{ mg}\cdot\text{kg}^{-1}$. IR thermal images (C) and tumor temperature evolutions (D) of orthotopic EMT-6 tumor-bearing mice after intravenous injection of PBS or $5 \text{ mg}\cdot\text{kg}^{-1} \text{ NP}_{\text{PPBBT}}$ under 808 nm laser irradiation at 0.5 or $1.0 \text{ W}/\text{cm}^2$ for 5 min.

Current Figure S23:

Figure S23. *Ex vivo* DiD-fluorescent images of the major organs (heart (H), liver (Li), spleen (S), lung (Lu), kidney (K), and tumor (T)) excised from orthotopic EMT-6 tumor-bearing BALB/c mice at 12, 24, 48, or 72 h post *i.v.* injection of $\text{NP}_{\text{PPBBT}}/\text{DiD}$ at a DiD dose of $0.25 \text{ mg}\cdot\text{kg}^{-1}$ for each mouse ($n = 3$ for each time point).

[Moreover, the saline injection only group should be added as a blank control, which aims to show if the laser alone ($1 \text{ w}/\text{cm}^2$) or nanoparticle alone would affect the tumor growth.]

We thank the reviewer for his/her instructive comments. Following his/her instruction, in current revision, using orthotopic breast tumor-bearing mice, we reconducted the *in vivo* tumor regression study of NP_{PPBBT} nanopartilcs and added one group of phosphate-buffered saline (PBS) injection only as the blank control. As shown in current Figure 5B, none of the three control groups (*i.e.*, $5 \text{ mg}\cdot\text{kg}^{-1} \text{ NP}_{\text{PPBBT}}$ without laser irradiation, PBS with $1.0 \text{ W}/\text{cm}^2$ laser irradiation, and PBS without laser irradiation) delayed the tumor growth, suggesting neither the laser alone nor NP_{PPBBT} nanopartilcs alone could inhibit tumor growth. We revised accordingly.

Original:

Page 14, line 7: Mice bearing EMT-6 tumor ~~xenografts~~ were randomly divided into ~~four~~ groups ($n = 5$ per group) among which three groups of mice were intravenously (*i.v.*) injected with 5

$\text{mg}\cdot\text{kg}^{-1}$ NP_{PPBBT} while ~~one group~~ of mice were *i.v.* injected with PBS. ~~Twelve hours~~ post injection, tumors of two experimental groups of NP_{PPBBT} -injected mice were irradiated with 808 nm laser for 10 min at a power density of 0.5 or 1.0 W/cm^2 , respectively. Tumors of one group of NP_{PPBBT} -injected mice without laser irradiation ~~and~~ tumors of PBS-injected group mice ~~under~~ 1.0 W/cm^2 laser irradiation were designated as ~~two~~ control groups. During the treatment, body weights of the mice were monitored and the results indicated that none of above ~~four~~-type treatments induced toxicity to the mice (Figure 5A). Time course tumor volume curves indicated that the tumor growth in the experimental group of $5 \text{ mg}\cdot\text{kg}^{-1}$ NP_{PPBBT} plus 1.0 W/cm^2 NIR laser irradiation was the most significantly inhibited (Figure 5B). Lower laser density (0.5 W/cm^2) together with $5 \text{ mg}\cdot\text{kg}^{-1}$ NP_{PPBBT} in the experimental group also showed significant inhibition on tumor growth but not as efficient as that of mice treated by higher laser density (1.0 W/cm^2). However, ~~neither~~ of the ~~two~~ control groups (*i.e.*, $5 \text{ mg}\cdot\text{kg}^{-1}$ NP_{PPBBT} without laser irradiation ~~and~~ PBS with 1.0 W/cm^2 laser irradiation) delayed the tumor growth (Figure 5B). After monitoring the body weights and tumor volumes of the mice for ~~18~~ 48 days, we sacrificed the mice, took photographs of the tumors (Figure 5C), and weighted the tumors (Figure 5D). The results indicated that the tumor growth in the experimental group of $5 \text{ mg}\cdot\text{kg}^{-1}$ NP_{PPBBT} plus 1.0 W/cm^2 laser irradiation was the mostly inhibited (Figure 5C-5D) while that between ~~two~~ control groups was not significant (Figure 5C-5D).

Revision:

Page 15, line 11: Mice bearing orthotopic EMT-6 tumor were randomly divided into five groups ($n = 5$ per group) among which three groups of mice were intravenously (*i.v.*) injected with $5 \text{ mg}\cdot\text{kg}^{-1}$ NP_{PPBBT} while two groups of mice were *i.v.* injected with PBS. Twenty-four hours post injection, tumors of two experimental groups of NP_{PPBBT} -injected mice were irradiated with 808 nm laser for 10 min at a power density of 0.5 or 1.0 W/cm^2 , respectively. Tumors of one group of NP_{PPBBT} -injected mice without laser irradiation and tumors of two groups of PBS-injected group mice with (or w/o) 1.0 W/cm^2 laser irradiation were designated as three control groups. During the treatment, body weights of the mice were monitored and the results indicated that none of above five-type treatments induced toxicity to the mice (Figure 5A). Time course tumor volume curves indicated that the tumor growth in the experimental group of $5 \text{ mg}\cdot\text{kg}^{-1}$ NP_{PPBBT} plus 1.0 W/cm^2 NIR laser irradiation was the most significantly inhibited (Figure 5B). Lower laser density (0.5 W/cm^2) together with $5 \text{ mg}\cdot\text{kg}^{-1}$ NP_{PPBBT} in the experimental group also showed significant inhibition on tumor growth but not as efficient as that of mice treated by higher laser density (1.0 W/cm^2). However, none of the three control groups (*i.e.*, $5 \text{ mg}\cdot\text{kg}^{-1}$ NP_{PPBBT} without laser irradiation, PBS with 1.0 W/cm^2 laser irradiation, and PBS without laser irradiation) delayed the tumor growth (Figure 5B), suggesting neither the laser alone nor NP_{PPBBT} nanoparticles alone could inhibit tumor growth. After monitoring

the body weights and tumor volumes of the mice for 15 days, we sacrificed the mice, took photographs of the tumors (Figure 5C), and weighted the tumors (Figure 5D). The results indicated that the tumor growth in the experimental group of $5 \text{ mg} \cdot \text{kg}^{-1} \text{ NP}_{\text{PPBBT}}$ plus 1.0 W/cm^2 laser irradiation was the mostly inhibited (Figure 5C-5D) while that among three control groups was not significant (Figure 5C-5D).

Original Figure 5

Figure 5 PTT efficiency of NP_{PPBBT} nanoparticles for anti-tumor treatment. Body weight (A) and tumor volume (B) curves of EMT-6 tumor-bearing mice at different time points after receiving one dose of treatment with indicated formulations. Photographs (C) and weights (D) of tumors in mice sacrificed after 18-day observation. (E) H&E, Ki67, and TUNEL analyses of the tumor tissues from mice sacrificed at day 18 post treatment. Scale bar, 50 μm . Results are presented as mean \pm SD; $n = 5$; * $p < 0.05$; ** $p < 0.01$; *** $p < 0.001$, analyzed by Student's t test.

Figure 5 PTT efficiency of NP_{PPBET} nanoparticles for anti-tumor treatment. Body weight (A) and tumor volume (B) curves of orthotopic EMT-6 tumor-bearing mice at different time points after receiving one dose of treatment with indicated formulations. At 24 h post injection, tumors were irradiated with (or w/o) laser for 10 min and the observations started. Photographs (C) and weights (D) of tumors in mice sacrificed after 15-day observation. (E) H&E staining, Ki67 (Cy3, red) and DAPI (blue) counterstaining, and TUNEL (FITC, green) and DAPI (blue) counterstaining of the tumor tissues from mice sacrificed at day 15 post treatment. Scale bars, 50 μm for H&E staining; 20 μm for Ki67 and TUNEL staining. Results are presented as mean ± SD; n = 5; **p* < 0.05; ***p* < 0.01; ****p* < 0.001, analyzed by Student's *t* test.

Reviewer #2

[In this manuscript, Peiyao, Yinchu, Zhen and coauthors applied the concept of the rapid condensation reaction between 2-cyano-benzothiazole (CBT) and the 1,2-aminothiols group of cysteine for design and facile synthesis of a new conjugated polymer poly-2-phenyl-benzobisthiazole (PPBBT). They characterized PPBBT in a series of studies in vitro, showed its comparable photothermal properties to existing conjugated polymers, and further generated PPBBT nanoparticles (NPPBBT) for photothermal treatment of a human breast cancer xenograft.

Before this study, many conjugated polymers with good photothermal properties have been reported and utilized for photothermal therapy of cancer. Nevertheless, the rapid CBT-Cys condensation reaction which enables facile synthesis and provides great potential for further modification has never been utilized to design and synthesize photothermal materials. This study is interesting and overall well structured. However, the lack of some key data in terms of the polymer synthesis remains the major flaw as described in detail below. Together, I recommend a major revision for this manuscript.]

We thank the reviewer for his/her positive assessment on this work. We agree with the reviewer the rapid CBT-Cys condensation reaction has never been utilized to design and synthesize photothermal materials, which addresses the novelty of this work. After addressing the major issues raised by the reviewer, we do think current revision could be considered for publication in *Nature Communications*.

[1. Reference 10 is not for photothermal therapy of cancer. May not be proper to cite here, but a minor issue.]

We thank the reviewer for him/her pointing out such an important issue. In current version, we replaced original Ref. 10 with another one which is more relevant to photothermal therapy of cancer and revised the manuscript accordingly. Please check.

Original reference 10:

10 ~~Lyu, Y., et al. Semiconducting Polymer Nanobioconjugates for Targeted Photothermal Activation of Neurons. *J. Am. Chem. Soc.* **138**, 9049-9052 (2016).~~

Revised reference 10:

10 Yang, Z. et al. Near-Infrared Semiconducting Polymer Brush and pH/GSH-Responsive Polyoxometalate Cluster Hybrid Platform for Enhanced Tumor-Specific Phototheranostics. *Angew. Chem. Int. Ed.* **57**, 14101-14105 (2018).

[2. NMR spectra of PPBBT and its analogs are required to confirm the chemical structures.]

We thank the reviewer for his/her instructive suggestions. Following his/her instructions, in current revision, we conducted ^1H NMR of **PPBBT** and supplemented the spectrum to the Supplementary Information. We failed to obtain the ^1H NMR spectrum of the analog **Dihexyl-PPBBT** due to its very poor solubility in common deuterated solvents. However, as indicated in our original submission, XPS data confirmed the formation of the C-N-C structures on the benzobisthiazole structure in both **PPBBT** and **Dihexyl-PPBBT**. We revised the manuscript accordingly.

Revisions:

Page 7, line 6: The crude product was then centrifuged, washed with water and ethanol to remove the catalyst and remaining starting materials, and lyophilized to yield pure black powder product (Figure 1B) and characterized with ^1H NMR spectrum (Supplementary Figure S1).

Page S2, line 7 from the bottom: ^1H NMR spectrum of **PPBBT** was obtained on a Bruker AV-400 MHz spectrometer.

Current Figure S1:

cpv-pbt-1221-3-41

Figure S1. ^1H NMR spectrum of **PPBBT** in $\text{DMSO-}d_6$.

[3. Gel Permeation Chromatography to measure the molecular weight or distribution of PPBBT and its analogs are required.]

We thank the reviewer for his/her very instructive comments. Following his/her instructions, in current revision, we conducted gel permeation chromatography (GPC) of **PPBBT**. As shown in current Figure S4 and Table S2, the number-average molecular weight of **PPBBT** was measured to be 23,176 with a polydispersity (PDI) of 1.34. We failed to obtain the GPC trace of **Dihexyl-PPBBT** due to its very poor solubility in common GPC eluents. We revised the manuscript accordingly. Please check.

Revisions:

Page 7, line 4 from the bottom: Additionally, the gel permeation chromatography (GPC) trace of **PPBBT** showed that the number-average molecular weight of the polymer was about 23,176 with a polydispersity (PDI) of 1.34 (Supplementary Figure S4 and Table S2).

Page S2, line 6 from the bottom: Molecular weight and molecular weight distribution of the polymer **PPBBT** with low solubility at room temperature were determined by gel permeation chromatography (GPC) with a PL 210 equipped with one Shodex AT-803S and two Shodex AT-806MS columns at 150 °C using trichlorobenzene as the eluent and calibrated with polystyrene standards.

Current Figure S4:

Figure S4. GPC trace of **PPBBT**.

Current Table S2:

Table S2. GPC trace analysis of **PPBBT** in Figure S4.

MW Averages

Peak No	Mp	Mn	Mw	Mz	Mz + 1	Mv	PDI
1	28,986	23,176	31,089	41,496	52,796	29,717	1.34

Processed Peaks

Peak No	Start RT (mins)	Max RT (mins)	End RT (mins)	Pk Height (mV)	%Height	Area (mV.secs)	%Area
1	13.12	14.15	15.02	-1.99685	100	125.849	100

[4. Absorption coefficient of the polymers at the excitation wavelength, and comparison to existing polymers or carbon tubes need to be presented.]

We thank the reviewer for his/her instructive comments. Following his/her instruction, in current revision, we measured the extinction coefficient of **NP_{PPBTT}** at 808 nm. As shown in current Figure S18, the extinction coefficient of **NP_{PPBTT}** in PBS at 808 nm was calculated to be $7.03 \text{ L}\cdot\text{g}^{-1}\cdot\text{cm}^{-1}$, which was slightly lower than that $20.2 \text{ L}\cdot\text{g}^{-1}\cdot\text{cm}^{-1}$ of polypyrrole nanosheet recently reported. We revised the manuscript accordingly.

Revision:

Page 11, line 7: The extinction coefficient of **NP_{PPBTT}** in PBS at 808 nm was calculated to be $7.03 \text{ L}\cdot\text{g}^{-1}\cdot\text{cm}^{-1}$ (Supplementary Figure S18), slightly lower than that $20.2 \text{ L}\cdot\text{g}^{-1}\cdot\text{cm}^{-1}$ of polypyrrole nanosheet recently reported³⁰.

30 Wang, X. *et al.* Ultrathin Polypyrrole Nanosheets via Space-Confined Synthesis for Efficient Photothermal Therapy in the Second Near-Infrared Window. *Nano Lett.* **18**, 2217-2225 (2018).

Current Figure S18:

Figure S18. (A) Vis-NIR spectra of **NP_{PPBTT}** in PBS at different concentrations. (B) Fitted calibration curve of Vis-NIR absorbance vs. concentration of **NP_{PPBTT}** in PBS at 808 nm.

[5. Mass data for Scheme S1 are required.]

We thank the reviewer for his/her instructive comments. According to his/her instruction, we supplemented the mass date of the product 2,5-dihexyl-1,4-dicyanobenzene in Scheme S1 to current Supplementary Information. Please check.

Original:

Page 7, line 9: To test the feasibility of introducing additional functional groups onto **PPBBT**, we synthesized 2,5-dihexyl-1,4-dicyanobenzene²⁶ (Scheme S1 and Supplementary Figures S3-S4) and used it to react with 2,5-dimercapto-1,4-phenylenediamine under the same reaction condition for the preparation of dihexyl-substituted **PPBBT** (**Dihexyl-PPBBT**, Figure 1A).

Revision:

Page 7, the last line: To test the feasibility of introducing additional functional groups onto **PPBBT**, we synthesized 2,5-dihexyl-1,4-dicyanobenzene²⁸ (Scheme S1 and Supplementary Figures S5-S7) and used it to react with 2,5-dimercapto-1,4-phenylenediamine under the same reaction condition for the preparation of dihexyl-substituted **PPBBT** (**Dihexyl-PPBBT**, Figure 1A).

Revision:

Page S2, line 2 from the bottom: The electron ionization-mass spectrometry (EI-MS) spectrum of 2,5-dihexyl-1,4-dicyanobenzene was obtained on a Thermo Scientific™ Q Exactive™ GC Orbitrap™ GC-MS/MS equipped with a standard EI (70 eV) source.

Current Figure S7:

Figure S7. EI-MS spectrum of 2,5-dihexyl-1,4-dicyanobenzene.

[6. Compared to PCB1 [photothermal conversion efficiency: 42.8% (Ref 33)] and PBIBDF-BF nanoparticles [photothermal conversion efficiency: 46.7% (Ref 32)], the photothermal conversion efficiency of NPPPBBT reported here is 32.4%. The authors need to justify the advantage of using NPPPBBT for photothermal therapy of cancer instead of PCB1 and PBIBDF-BF.]

We thank the reviewer for pointing out such important issue. Compared to other conjugated polymers, the biggest advantage of our conjugated polymers lies in their facile syntheses. The reaction used for their syntheses is efficient and the reaction condition is mild. The starting materials for their syntheses are inexpensive and feasible for modification. Although the photothermal conversion efficiency of our polymer was lower than those of PCB 1 (current Ref 36) and PBIBDF-BF (current Ref 35) reported, synthesis of our polymer was much more mild and efficient. Please check.

[7. For *in vitro* photo cytotoxicity study, the uptake of NPPPBBT by EMT6 cells need to be determined.]

We thank the reviewer for his/her instructive comments. According to his/her instruction, in current revision, we studied the cell uptake of NP_{PPBBT} by EMT-6 cells using DiD as the fluorescence indicator. As shown in current Figure S21, the red fluorescence of DiD gradually increased inside cells and approached its plateau at 4 h, suggesting a fairly good cellular uptake of the nanoparticles by cells in 4 h. We revised the manuscript accordingly.

Original:

Page 11, line 2 from the bottom: **Photo cytotoxicity of NP_{PPBBT} nanoparticles.** With their good *in vitro* photothermal property verified, NP_{PPBBT} nanoparticles were subsequently applied for *in vivo* PTT of tumor. Before that, photo cytotoxicity of NP_{PPBBT} nanoparticles was evaluated on EMT-6 cells with 3-(4,5-dimethylthiazol-2-yl) 2,5 diphenyltetrazolium bromide (MTT) assay.

Revision:

Page 12, line 11: **Cell uptake and photo cytotoxicity of NP_{PPBBT} nanoparticles.** With their good *in vitro* photothermal property verified, NP_{PPBBT} nanoparticles were subsequently applied for *in vivo* PTT of tumor. Before that, 1,1'-dioctadecyl-3,3,3',3'-tetramethylindodicarbocyanine, 4-chlorobenzenesulfonate salt (DiD), a NIR fluorescent dye, was co-encapsulated with PPBBT to form NP_{PPBBT}/DiD nanoparticles to study the cell uptake property of NP_{PPBBT} nanoparticles by EMT-6 breast cells. NP_{PPBBT}/DiD nanoparticles were prepared using the same

nanoprecipitation method as that of NP_{PPBBT} nanoparticles and the loading efficiency of DiD was calculated to be 83% (Supplementary Figure S20). Confocal laser scanning microscopy fluorescence images of the cells incubated with NP_{PPBBT}/DiD for 0.25, 0.5, 1, 2, 4, or 8 h showed that the red DiD fluorescence gradually increased inside cells and approached its plateau at about 4 h³⁷ (Supplementary Figure S21). Whereafter, photo cytotoxicity of NP_{PPBBT} nanoparticles on EMT-6 cells after 4 h incubation with the nanoparticles was evaluated using 3-(4,5-dimethylthiazol-2-yl) 2,5 diphenyltetrazolium bromide (MTT) assay.

37 Sun, C. *et al.* Effect of hydrophobicity of core on the anticancer efficiency of micelles as drug delivery carriers. *ACS Appl. Mater. Interfaces* **6**, 22709-22718 (2014).

Revision:

Page S4, line 8 from the bottom: **Cell uptake of nanoparticles.** At first, the NP_{PPBBT}/DiD was prepared. mPEG-*b*-PHEP (10.0 mg), 1,1'-dioctadecyl-3,3,3',3'-tetramethylindodicarbocyanine, 4-chlorobenzenesulfonate salt (DiD 0.025 mg) and PPBBT (1.0 mg) was dissolved in 2.0 mL DMSO, and then the solution was dropwise added into 10 mL ultrapure water under stirring. After stirring for 2 h, the solution was dialyzed against (MWCO: 14000 Da) ultrapure water for 24 h to remove DMSO. The EMT-6 breast cells were seeded on coverslips in 24-well plates overnight. The medium was replaced with fresh complete medium containing NP_{PPBBT} nanoparticles (50.0 µg/mL). After being incubated for 0.25, 0.5, 1, 2, 4, or 8 h at 37 °C, the cells were washed with PBS, fixed with 1% formaldehyde, and counterstained with Alexa Fluor 488 phalloidin (Invitrogen, Thermo Fisher) and 4, 6-diamidino-2-phenylindole (DAPI, Sigma-Aldrich). Finally, the slices were observed by Zeiss LSM 880 confocal laser scanning microscope (Carl Zeiss, Oberkochen, Germany).

Current Figure S20:

Figure S20. Determination of the loading efficiency of DiD in NP_{PPBBT}/DiD. (A) Fluorescent spectra of DiD dissolved in DMSO at different concentrations. (B) Fitted calibration curve of fluorescence intensity vs. concentration of DiD in DMSO at 670 nm. (C) Fluorescent spectrum of NP_{PPBBT}/DiD dissolved in DMSO (diluted 100 times). The concentration of DiD in the NP_{PPBBT}/DiD solution was calculated to be 0.207 µg/mL, thus the loading efficiency of DiD in

NP_{PPBBT}/DiD was about 83% $\{(0.207 \mu\text{g}/\text{mL} * 100) / 0.025 \text{ mg}/\text{mL} * 100\% = 83\%$. Excitation: 633 nm.

Current Figure S21:

Figure S21. (A) Confocal laser scanning microscopy fluorescence images of EMT-6 breast cells after incubation with NP_{PPBBT}/DiD nanoparticles for 0.25, 0.5, 1, 2, 4, or 8 h and then washed with PBS. Cell nucleus and cytoskeleton were stained by DAPI (blue) and Alexa Fluor 488 phalloidin (green), respectively. (B) Time course geometric mean fluorescence intensity (GMFI) of DiD in A.

[From Method, the cell medium with particles was not replaced after 4 hours incubation, raising the question whether the particles were taken into cells or just on cell surface.]

We thank the reviewer for him/her pointing out such an important issue. Actually, the cells were washed with PBS for three times and replaced with fresh medium before being exposed to NIR laser. We are sorry for not clearly stating this in previous submission. In current revision, we clearly stated the cell experiment protocol in the Supplementary Information. Please check.

Revision:

Page S5, line 9: After incubation for 4 h, the cells were washed with PBS for three times, added with fresh medium, and then exposed to the NIR laser (808 nm, 1.0 W/cm², 10 min) and further incubated for 48 h, and then the cell viability was analyzed by MTT assay.

[8. For ex vivo IHC staining of tumor slides in figure 5E: First of all, the resolution is relatively low; second, I understand H&E, part of the TUNEL staining, but not Ki67. For TUNEL, I think the blue may be DAPI and the green may be the double strand DNA breaks. It needs description. For Ki67, if the blue indicates nucleus and brown staining indicate Ki67 protein, why PBS-IR group showed significantly less nucleus but stronger proliferation staining?]

We thank the reviewer for his/her instructive comments. To address the reviewer's questions, in current revision, we acquired new images with higher resolution (current Figure S27E). As we know, TUNEL is a method for detecting DNA fragmentation by labeling the 3'-hydroxyl termini in the double-strand DNA breaks generated during apoptosis. And the Ki67 protein is a cellular marker which is strictly associated with cell proliferation. During interphase, the Ki67 antigen can be exclusively detected within the cell nucleus, whereas in mitosis most of the protein is relocated to the surface of the chromosomes. Ki67 protein is present during all active phases of the cell cycle (G₁, S, G₂, and mitosis), but is absent in resting (quiescent) cells (G₀). In our experiment, for TUNEL, the blue fluorescence was DAPI and the green was TUNEL. And for Ki67, **the Ki67-positive nuclei were brown** and Ki67-negative nuclei were blue. **In current version**, we further conducted the animal experiments using orthotopic breast tumor mode and **used immunofluorescence to track Ki67 (red)** to let Ki67 staining more distinct. As shown in the second row in current Figure 5E, the blue fluorescence was DAPI and the red was Ki67. Moreover, in current version, H&E staining of the major organ tissues were also conducted to analyze cell state in all major organs of the mice sacrificed at day 15 post treatment to confirm that **NP_{PPBBT}** nanoparticles are biocompatible. All these results suggested that our **NP_{PPBBT}** nanoparticles are suitable for PTT of tumors *in vivo*, pathologically and morphologically. We revised the manuscript accordingly.

Origin:

Page 15, line 7 from the bottom: Furthermore, hematoxylin-eosin (H&E) staining, Ki67 ~~antigen~~ staining, and terminal transferased UTP nick-end labeling (TUNEL) assay of the tumor tissues to analyze cell state, cell proliferation, and cell apoptosis in the tissues, respectively, were conducted after PTT treatment (Figure 5E).

Revision:

Page 16, line 3 from the bottom: Furthermore, hematoxylin-eosin (H&E) staining, Ki67 immunofluorescence staining³⁹, and terminal transferased UTP nick-end labeling (TUNEL)

assay of the tumor tissues to analyze cell state, cell proliferation, and cell apoptosis in the tissues, respectively, were conducted after PTT treatment (Figure 5E).

39 Solomon, K. R. *et al.* Ezetimibe Is an Inhibitor of Tumor Angiogenesis. *Am. J. Pathol.* **174**, 1017-1026 (2009).

Revision:

Page 17, line 5: Moreover, H&E staining of the major organ tissues of the mice sacrificed at day 15 post treatment indicated that there was no obvious damage in all major organs, suggesting that NP_{PPBBT} nanoparticles were biocompatible (Supplementary Figure S24).

Original Figure 5:

Figure 5 PTT efficiency of NP_{PPBBT} nanoparticles for anti-tumor treatment. Body weight (A) and tumor volume (B) curves of EMT-6 tumor-bearing mice at different time points after receiving one dose of treatment with indicated formulations. Photographs (C) and weights (D) of tumors in

mice sacrificed after 18-day observation. (E) H&E, Ki67, and TUNEL analyses of the tumor tissues from mice sacrificed at day 18 post treatment. Scale bar, 50 μm . Results are presented as mean \pm SD; $n = 5$; $*p < 0.05$; $**p < 0.01$; $***p < 0.001$, analyzed by Student's t test.

Current Figure S27:

Figure S27. PTT efficiency of NP_{PBBT} nanoparticles for anti-tumor treatment. Body weight (A) and tumor volume (B) curves of subcutaneous EMT-6 tumor-bearing mice at different time points after receiving one dose of treatment with indicated formulations. At 12 h post injection, tumors were irradiated with (or w/o) laser for 10 min and the observations started. Photographs (C) and weights (D) of tumors in mice sacrificed after 18-day observation. (E) H&E staining, Ki67 staining (brown: Ki67-positive nuclei, blue: Ki67-negative nuclei), and TUNEL (FITC, green) and DAPI (blue) counterstaining of the tumor tissues from mice sacrificed at day 18 post treatment. Scale bar, 50 μm . Results are presented as mean \pm SD; $n = 5$; $*p < 0.05$; $**p < 0.01$; $***p < 0.001$, analyzed by Student's t test.

Revised Figure 5:

Figure 5 PTT efficiency of NP_{PPBBT} nanoparticles for anti-tumor treatment. Body weight (A) and tumor volume (B) curves of orthotopic EMT-6 tumor-bearing mice at different time points after receiving one dose of treatment with indicated formulations. At 24 h post injection, tumors were irradiated with (or w/o) laser for 10 min and the observations started. Photographs (C) and weights (D) of tumors in mice sacrificed after 15-day observation. (E) H&E staining, Ki67 (Cy3, red) and DAPI (blue) counterstaining, and TUNEL (FITC, green) and DAPI (blue) counterstaining of the tumor tissues from mice sacrificed at day 15 post treatment. Scale bars, 50 μ m for H&E staining; 20 μ m for Ki67 and TUNEL staining. Results are presented as mean \pm SD; n = 5; * p < 0.05; ** p < 0.01; *** p < 0.001, analyzed by Student's t test.

Current Figure S24:

Figure S24. H&E staining of the major organs (heart, liver, spleen, lung, and kidney) of mice sacrificed at day 15 post treatment. Scale bar, 50 μ m.

Reviewer #3

[The manuscript “Facile syntheses of conjugated polymers for photothermal therapy of tumor” by Chen et al developed a new conjugated polymer with facile method and systematically explore its therapeutic efficacy. This study was well designed and there are convincing evidences to support the conclusions. Therefore, I recommend the acceptance of this manuscript after minor revision.]

We thank the reviewer for his/her very positive assessment on this work. We totally agreed with the reviewer that our study was well designed and there are convincing evidences to support the conclusions. After addressing the major issue raised by the reviewer, we do think current revised manuscript could be considered for publication in *Nature Communications*.

[1. In this study, authors used subcutaneous breast tumor model to systematically evaluate the therapeutic efficacy of the developed conjugated polymers. Actually, there are big differences between subcutaneous tumors and orthotopic ones. Therefore, I wonder whether authors could provide related data using orthotopic breast tumor model. This will greatly approve the unique advantages of the developed photothermal nanoplatform.]

We thank the reviewer for his/her instructive suggestions. According to the reviewer's suggestion, in current revision, we used orthotopic breast tumor model to further evaluate the PTT efficiency of our **NP_{PPBBT}** nanoparticles. As shown in current Figure 4 and Figure 5, **NP_{PPBBT}** nanoparticles also exhibited excellent antitumor efficiency for orthotopic breast tumors as well. We revised the manuscript accordingly.

Original:

Page 12, line 4 from the bottom: We then investigated the pharmacokinetics and ~~biodistribution~~ of **NP_{PPBBT}** nanoparticles in EMT-6 breast tumor-bearing mice. ~~1,1'-dioctadecyl-3,3,3',3'-tetramethylindodicarbocyanine, 4-chlorobenzenesulfonate salt (DiD), a NIR fluorescent dye, was used to track the biodistribution of **NP_{PPBBT}** nanoparticles *in vivo*. **NP_{PPBBT}/DiD** was prepared using the same nanoprecipitation method as that of **NP_{PPBBT}** and intravenously (*i.v.*) injected into the tumor-bearing mice at a DiD concentration of 1.25 mg•kg⁻¹. Blood drug concentration-time curve of **NP_{PPBBT}/DiD** indicated that the elimination half life ($t_{1/2}$) of **NP_{PPBBT}** was about 10.32 h (Figure 4A), which was comparable to those of other reported PTT materials^{34,35}. To investigate the biodistribution of **NP_{PPBBT}**, the tumor-bearing mice were sacrificed at 6, 12, or 24 h post *i.v.* injection of **NP_{PPBBT}/DiD** and the major organs (heart, liver, spleen, lung, kidney, and tumor) were imaged using an Xenogen IVIS® spectrum system. As~~

shown in Figure 4B and Supplementary Figure S16, DiD fluorescence imaging of the organs indicated that the NP_{PPBBT} nanoparticles were mainly located in the liver followed by the tumor. However, the fluorescence of DiD in tumor increased with time, indicating the effective accumulation of the NP_{PPBBT} nanoparticles in the tumor site due to the EPR effect, which was favorable for cancer treatment. Accordingly, EMT-6 tumor-bearing BALB/c nude mice were *i.v.* injected with NP_{PPBBT} nanoparticles at a dose of 5 mg•kg⁻¹. Twelve hours post-injection, the tumors in mice were irradiated with an 808 nm laser for 5 min at a power density of 0.5 or 1.0 W/cm². The spatial temperature distribution and the temperature increase at the tumor sites were real-time monitored with an IR imaging camera (Figure 4C). As shown in Figure 4D, the temperature of the tumor region of the NP_{PPBBT}-injected group raised quickly by 7.5 °C within 3 min under 0.5 W/cm² laser irradiation and 16.7 °C within 2 min under 1.0 W/cm² laser irradiation, respectively.

Revision:

Page 13, the last line: We then investigated the pharmacokinetics of NP_{PPBBT} nanoparticles in healthy mice and orthotopic EMT-6 tumor-bearing mice. NP_{PPBBT}/DiD was intravenously (*i.v.*) injected into the healthy mice or tumor-bearing mice at a DiD concentration of 0.25 mg•kg⁻¹. Blood drug concentration-time curves of NP_{PPBBT}/DiD indicated that the elimination half life (T_{1/2z}) of NP_{PPBBT} was about 95.4 h in healthy mice and 46.6 h in tumor-bearing mice (Figure 4A and Supplementary Table S4) according to the noncompartment analysis (DAS 3.2.6). The long circulation property of NP_{PPBBT}/DiD might lead to its efficient accumulation in tumors due to its EPR effect³⁸, evidenced by that its T_{1/2z} of 95.4 h in healthy mice was shortened to 46.6 h in tumor-bearing mice. To investigate the biodistribution of NP_{PPBBT}, the tumor-bearing mice were sacrificed at 12, 24, 48, or 72 h post *i.v.* injection of NP_{PPBBT}/DiD and the major organs (heart, liver, spleen, lung, kidney, and tumor) were imaged using an Xenogen IVIS® spectrum system. Both the biodistribution data in Figure 4B and the *ex vivo* fluorescence images in Supplementary Figure S23 of the major organs consistently indicated that, tumors had the highest average DiD fluorescence among all the major organs studied and their fluorescence reached to its maximum at 24 h then gradually decreased. Above results indicated that, due to tumor EPR effect, our NP_{PPBBT} nanoparticles were effectively accumulated in tumors which was favorable for their PTT on tumors. Accordingly, orthotopic EMT-6 tumor-bearing BALB/c mice were *i.v.* injected with NP_{PPBBT} nanoparticles at a dose of 5 mg•kg⁻¹. Twenty-four hours post injection, the tumors in mice were irradiated with an 808 nm laser for 5 min at a power density of 0.5 or 1.0 W/cm². The spatial temperature distribution and the temperature increase at the tumor sites were real-time monitored with an IR imaging camera (Figure 4C). As shown in Figure 4D, the temperature of the tumor region of the NP_{PPBBT}-injected group raised quickly

by 5.2 °C within 1 min under 0.5 W/cm² laser irradiation and 16.5 °C within 2 min under 1.0 W/cm² laser irradiation, respectively.

38 Xu, X., Ho, W., Zhang, X., Bertrand, N. & Farokhzad, O. Cancer nanomedicine: from targeted delivery to combination therapy. *Trends Mol. Med.* **21**, 223-232 (2015).

Original:

Page 14, line 7: Mice bearing EMT-6 tumor ~~xenografts~~ were randomly divided into ~~four~~ groups (n = 5 per group) among which three groups of mice were intravenously (*i.v.*) injected with 5 mg•kg⁻¹ NP_{PPBBT} while ~~one group~~ of mice were *i.v.* injected with PBS. ~~Twelve~~ hours post injection, tumors of two experimental groups of NP_{PPBBT}-injected mice were irradiated with 808 nm laser for 10 min at a power density of 0.5 or 1.0 W/cm², respectively. Tumors of one group of NP_{PPBBT}-injected mice without laser irradiation ~~and~~ tumors of PBS-injected group mice ~~under~~ 1.0 W/cm² laser irradiation were designated as ~~two~~ control groups. During the treatment, body weights of the mice were monitored and the results indicated that none of above ~~four~~-type treatments induced toxicity to the mice (Figure 5A). Time course tumor volume curves indicated that the tumor growth in the experimental group of 5 mg•kg⁻¹ NP_{PPBBT} plus 1.0 W/cm² NIR laser irradiation was the most significantly inhibited (Figure 5B). Lower laser density (0.5 W/cm²) together with 5 mg•kg⁻¹ NP_{PPBBT} in the experimental group also showed significant inhibition on tumor growth but not as efficient as that of mice treated by higher laser density (1.0 W/cm²). However, ~~neither~~ of the ~~two~~ control groups (*i.e.*, 5 mg•kg⁻¹ NP_{PPBBT} without laser irradiation ~~and~~ PBS with 1.0 W/cm² laser irradiation) delayed the tumor growth (Figure 5B). After monitoring the body weights and tumor volumes of the mice for ~~18~~ days, we sacrificed the mice, took photographs of the tumors (Figure 5C), and weighted the tumors (Figure 5D). The results indicated that the tumor growth in the experimental group of 5 mg•kg⁻¹ NP_{PPBBT} plus 1.0 W/cm² laser irradiation was the mostly inhibited (Figure 5C-5D) while that ~~between two~~ control groups was not significant (Figure 5C-5D). Above results indicated that our NP_{PPBBT} had high PTT efficiency for antitumor treatment. Furthermore, hematoxylin-eosin (H&E) staining, Ki67 ~~antigen~~ staining, and terminal transferase UTP nick-end labeling (TUNEL) assay of the tumor tissues to analyze cell state, cell proliferation, and cell apoptosis in the tissues, respectively, were conducted after PTT treatment (Figure 5E).

Revision:

Page 15, line 11: Mice bearing orthotopic EMT-6 tumor were randomly divided into five groups (n = 5 per group) among which three groups of mice were intravenously (*i.v.*) injected with 5 mg•kg⁻¹ NP_{PPBBT} while two groups of mice were *i.v.* injected with PBS. Twenty-four hours post injection, tumors of two experimental groups of NP_{PPBBT}-injected mice were irradiated with 808 nm laser for 10 min at a power density of 0.5 or 1.0 W/cm², respectively. Tumors of

one group of **NP_{PPBBT}**-injected mice without laser irradiation and tumors of two groups of PBS-injected group mice with (or w/o) 1.0 W/cm² laser irradiation were designated as three control groups. During the treatment, body weights of the mice were monitored and the results indicated that none of above five-type treatments induced toxicity to the mice (Figure 5A). Time course tumor volume curves indicated that the tumor growth in the experimental group of 5 mg•kg⁻¹ **NP_{PPBBT}** plus 1.0 W/cm² NIR laser irradiation was the most significantly inhibited (Figure 5B). Lower laser density (0.5 W/cm²) together with 5 mg•kg⁻¹ **NP_{PPBBT}** in the experimental group also showed significant inhibition on tumor growth but not as efficient as that of mice treated by higher laser density (1.0 W/cm²). However, none of the three control groups (i.e., 5 mg•kg⁻¹ **NP_{PPBBT}** without laser irradiation, PBS with 1.0 W/cm² laser irradiation, and PBS without laser irradiation) delayed the tumor growth (Figure 5B), suggesting neither the laser alone nor **NP_{PPBBT}** nanoparticles alone could inhibit tumor growth. After monitoring the body weights and tumor volumes of the mice for 15 days, we sacrificed the mice, took photographs of the tumors (Figure 5C), and weighted the tumors (Figure 5D). The results indicated that the tumor growth in the experimental group of 5 mg•kg⁻¹ **NP_{PPBBT}** plus 1.0 W/cm² laser irradiation was the mostly inhibited (Figure 5C-5D) while that among three control groups was not significant (Figure 5C-5D). Above results indicated that our **NP_{PPBBT}** had high PTT efficiency for antitumor treatment. Furthermore, hematoxylin-eosin (H&E) staining, Ki67 immunofluorescence staining³⁹, and terminal transferase UTP nick-end labeling (TUNEL) assay of the tumor tissues to analyze cell state, cell proliferation, and cell apoptosis in the tissues, respectively, were conducted after PTT treatment (Figure 5E).

39 Solomon, K. R. *et al.* Ezetimibe Is an Inhibitor of Tumor Angiogenesis. *Am. J. Pathol.* **174**, 1017-1026 (2009).

Revision:

Page 17, line 5: Moreover, H&E staining of the major organ tissues of the mice sacrificed at day 15 post treatment indicated that there was no obvious damage in all major organs, suggesting that **NP_{PPBBT}** nanoparticles were biocompatible (Supplementary Figure S24). Meanwhile, the pharmacokinetics, biodistribution, and photothermal performance on tumors of **NP_{PPBBT}** nanoparticles in subcutaneously EMT-6 tumor-bearing mice, as well as anti-tumor PTT efficiency of the nanoparticles on the tumor-bearing mice, were evaluated (Supplementary Figures S25-27). The results indicated that our **NP_{PPBBT}** nanoparticles also exhibited excellent antitumor efficiency on subcutaneous breast tumor as well.

Original Figure 4:

Figure 4 Pharmacokinetics, biodistribution, and photothermal performance in tumors of NP_{PPBTT} nanoparticles *in vivo*. (A) Blood DiD concentration vs. time curve in EMT-6 breast tumor-bearing mice intravenously injected with NP_{PPBTT}/DiD at a DiD dose of 1.25 mg•kg⁻¹ (n = 3 for each group). (B) Quantification of DiD fluorescence from the major organs (heart, liver, spleen, lung, kidney, and tumors) in tumor-bearing mice sacrificed at 6, 12, or 24 h post-injection of NP_{PPBTT}/DiD at a DiD concentration of 1.25 mg•kg⁻¹. IR thermal images (C) and tumor temperature evolutions (D) of EMT-6 tumor-bearing mice after intravenous injection of PBS or 5 mg•kg⁻¹ NP_{PPBTT} under 808 nm laser irradiation at 0.5 or 1.0 W/cm² for 5 min.

Revised Figure 4:

Figure 4 Pharmacokinetics, biodistribution, and photothermal performance in tumors of NP_{PPBBT} nanoparticles *in vivo*. (A) Blood DiD concentration vs. time curves in healthy mice or orthotopic EMT-6 breast tumor-bearing mice intravenously injected with NP_{PPBBT}/DiD at a DiD dose of 0.25 mg•kg⁻¹ (n = 3 for each group). (B) Quantification of DiD fluorescence from the major organs (heart, liver, spleen, lung, kidney, and tumors) in orthotopic EMT-6 tumor-bearing mice sacrificed at 12, 24, 48, or 72 h post injection of NP_{PPBBT}/DiD at a DiD concentration of 0.25 mg•kg⁻¹. IR thermal images (C) and tumor temperature evolutions (D) of orthotopic EMT-6 tumor-bearing mice after intravenous injection of PBS or 5 mg•kg⁻¹ NP_{PPBBT} under 808 nm laser irradiation at 0.5 or 1.0 W/cm² for 5 min.

Original Figure 5:

Figure 5 PTT efficiency of NP_{PPBBT} nanoparticles for anti-tumor treatment. Body weight (A) and tumor volume (B) curves of EMT-6 tumor-bearing mice at different time points after receiving one dose of treatment with indicated formulations. Photographs (C) and weights (D) of tumors in

mice sacrificed after 18-day observation. (E) H&E, Ki67, and TUNEL analyses of the tumor tissues from mice sacrificed at day 18 post treatment. Scale bar, 50 μm . Results are presented as mean \pm SD; n = 5; * $p < 0.05$; ** $p < 0.01$; *** $p < 0.001$, analyzed by Student's t test.

Revised Figure 5:

Figure 5 PTT efficiency of NP_{PPBBT} nanoparticles for anti-tumor treatment. Body weight (A) and tumor volume (B) curves of orthotopic EMT-6 tumor-bearing mice at different time points after receiving one dose of treatment with indicated formulations. At 24 h post injection, tumors were irradiated with (or w/o) laser for 10 min and the observations started. Photographs (C) and weights (D) of tumors in mice sacrificed after 15-day observation. (E) H&E staining, Ki67 (Cy3, red) and DAPI (blue) counterstaining, and TUNEL (FITC, green) and DAPI (blue) counterstaining of the tumor tissues from mice sacrificed at day 15 post treatment. Scale bars, 50 μm for H&E staining;

20 μm for Ki67 and TUNEL staining. Results are presented as mean \pm SD; $n = 5$; $*p < 0.05$; $**p < 0.01$; $***p < 0.001$, analyzed by Student's t test.

Current Figure S23:

Figure S23. *Ex vivo* DiD-fluorescent images of the major organs (heart (H), liver (Li), spleen (S), lung (Lu), kidney (K), and tumor (T)) excised from orthotopic EMT-6 tumor-bearing BALB/c mice at 12, 24, 48, or 72 h post *i.v.* injection of NP_{PPBBT}/DiD at a DiD dose of 0.25 mg•kg⁻¹ for each mouse (n = 3 for each time point).

Current Figure S24:

Figure S24. H&E staining of the major organs (heart, liver, spleen, lung, and kidney) of mice sacrificed at day 15 post treatment. Scale bar, 50 μm .

Current Figure S25:

Figure S25. Pharmacokinetics, biodistribution, and photothermal performance in tumors of NP_{PPBBT} nanoparticles *in vivo*. (A) Blood DiD concentration vs. time curve in subcutaneous EMT-6 breast tumor-bearing mice intravenously injected with NP_{PPBBT}/DiD at a DiD dose of $0.25 \text{ mg} \cdot \text{kg}^{-1}$. (B) Quantification of DiD fluorescence from the major organs (heart, liver, spleen, lung, kidney, and tumors) in tumor-bearing mice sacrificed at 6, 12, or 24 h post injection of NP_{PPBBT}/DiD at a DiD concentration of $0.25 \text{ mg} \cdot \text{kg}^{-1}$. IR thermal images (C) and tumor temperature evolutions (D) of subcutaneous EMT-6 tumor-bearing mice at 12 h post *i.v.* injection of PBS or $5 \text{ mg} \cdot \text{kg}^{-1}$ NP_{PPBBT} under 808 nm laser irradiation at 0.5 or 1.0 W/cm^2 for 5 min.

Current Figure S26:

Figure S26. *Ex vivo* DiD-fluorescent images of the major organs (heart (H), liver (Li), spleen (S), lung (Lu), kidney (K), and tumor (T)) excised from subcutaneous EMT-6 tumor-bearing BALB/c mice at 6, 12, or 24 h *post i.v.* injection of **NP_{PPBBT}/DiD** at a DiD dose of 0.25 mg•kg⁻¹ for each mouse (n = 3 for each time point).

Current Figure S27:

Figure S27. PTT efficiency of **NP_{PPBBT}** nanoparticles for anti-tumor treatment. Body weight (A) and tumor volume (B) curves of subcutaneous EMT-6 tumor-bearing mice at different time points after receiving one dose of treatment with indicated formulations. At 12 h post injection, tumors were irradiated with (or w/o) laser for 10 min and the observations started. Photographs (C) and weights (D) of tumors in mice sacrificed after 18-day observation. (E) H&E staining, Ki67 staining (brown: Ki67-positive nuclei, blue: Ki67-negative nuclei), and TUNEL (FITC, green) and DAPI (blue) counterstaining of the tumor tissues from mice sacrificed at day 18 post treatment. Scale bar, 50 μm.

Results are presented as mean \pm SD; n = 5; * p < 0.05; ** p < 0.01; *** p < 0.001, analyzed by Student's t test.

Current Table S4:

Table S4. Pharmacokinetic parameters of NP_{PPBBT}/DiD nanoparticles intravenously administered to the healthy mice or orthotopic EMT-6 tumor-bearing mice in Figure 4A. The data were obtained by noncompartement analysis (DAS 3.2.6).

Parameter	C _{max} (μg/mL)	T _{1/2z} (h)	AUC _{0-72 h} (μg/L•h)	CLz (L/h/kg)
Healthy mice	92.8	95.4	3131.6	0.716
Orthotopic EMT-6 tumor-bearing mice	88.6	46.6	2018.6	1.73

C_{max}, Peak concentrarion;
AUC, Area under the curve;

T_{1/2z}, Elimination half life;
CLz, Clearance rate.

[2. On page 13, line 14, it was mentioned that EMT-6 tumor-bearing nude mice were used in this study for the cancer treatment. I recommend authors to check whether it is nude mice or not.]

We thank the reviewer for pointing out this important issue. We are sorry for the mistake we made in the original submission. The mice used in the original submission were BALB/c mice but not nude mice. In current revision, we corrected this throughout the manuscript. Please check.

Original:

Page 13, line 6 from the bottom: Accordingly, EMT-6 tumor-bearing BALB/c ~~nude~~ mice were *i.v.* injected with NP_{PPBBT} nanopartilles at a dose of 5 mg•kg⁻¹.

Revision:

Page 14, line 2 from the bottom: Accordingly, orthotopic EMT-6 tumor-bearing BALB/c mice were *i.v.* injected with NP_{PPBBT} nanopartilles at a dose of 5 mg•kg⁻¹.

Original:

Page 15, line 5: After monitoring the body weights and tumor volumes of the ~~nude~~ mice for 18 days, we sacrificed the mice, took photographs of the tumors (Figure 5C), and weighted the tumors (Figure 5D).

Revision:

Page 16, line 8 from the bottom: After monitoring the body weights and tumor volumes of the mice for 15 days, we sacrificed the mice, took photographs of the tumors (Figure 5C), and weighted the tumors (Figure 5D).

Original:

Page 18, line 7: They performed syntheses, characterizations, photothermal studies and photothermal therapy of ~~nude~~ mice.

[3. Recently, accumulating evidences indicated that nanoplatfrom-based photothermal therapy could induce antitumor immune responses to combat tumor progression (Nat Commun, 2016, 7, DOI:10.1038/ncomms13193). In addition, it was evidenced that some conjugated polymers could serve as vaccine adjuvant to activate immune cells (Nanoscale, 2015, 7, 19282-19292). Therefore, I recommend authors to give more comprehensive discussions in the introduction.]

We thank the reviewer for his/her instructive suggestion. Following the reviewer's suggestion, we supplemented current manuscript with more comprehensive discussions on recent advances of conjugated polymers in the introduction section. Please check.

Revisions:

Page 3, the last line: Accumulating evidences indicated that nano-agent (e.g., CNT)-assisted PTT might bring anti-tumor immunological effects by generating tumor-associate agents from ablated tumor cell residues¹³, rendering PTT a powerful tool for cancer therapy.

Page 4, line 9: Moreover, some conjugated polymers were reported to interact with the immune system and might be used as potential vaccine adjuvants¹⁶.

13 Chen Q. et al. Photothermal therapy with immune-adjuvant nanoparticles together with checkpoint blockade for effective cancer immunotherapy. Nat. Commun. 7, 13193 (2016).

16 Gong H. et al. Stimulation of immune systems by conjugated polymers and their potential as an alternative vaccine adjuvant. Nanoscale 7, 19282-19292 (2015).

[1. On page 5, line 18, "...modified with additional functional groups..." should be "...modified with additional functional groups..."

2. On page 9, line 6, "To caculate the photothermal conversion efficiency, ..." should be "To calculate the photothermal conversion efficiency, ..." .]

We thank the reviewer for his/her careful reading. In current version, we carefully read through the manuscript and corrected corresponding typos. Please check.

Original:

Page 5, line 17: **PPBBT** could be easily modified with ~~additional~~ functional groups,

Revision:

Page 6, line 4: **PPBBT** could be easily modified with additional functional groups,

Original:

Page 9, line 6: To ~~eaaculate~~ the photothermal conversion efficiency,

Revision:

Page 9, line 4 from the bottom: To calculate the photothermal conversion efficiency,

REVIEWERS' COMMENTS:

Reviewer #1 (Remarks to the Author):

Title: Facile syntheses of conjugated polymers for photothermal therapy of tumor

Manuscript ID: NCOMMS-18-33848A

Authors: Peiyao Chen, Yinchu Ma, Zhen Zheng, Chengfan Wu, Yucai Wang, and Gaolin Liang

Comments:

1) I agree the method for synthesizing the conjugated polymer is novel and the conjugated polymer has never been applied for photothermal therapy, but "the first time" does not always mean "novel". Therefore, more comparison among other conjugated polymers (i.e. PANI, PEDOT, Ppy, et al.) should be conducted to highlight the uniqueness of the conjugated polymer in terms of photothermal therapy.

2) For the biodistribution assay, it might not be accurate to use the ex vivo imaging system to quantify the actual amounts of nanomaterials that have been accumulated in specific organs for the following reasons:

a: The self-quenching effect of fluorophore when the local concentration reached the threshold.

b: The thickness of the organ might affect the fluorescence intensity.

Therefore, it is highly recommended to measure the fluorescence of the homogenized tissue after dilutions.

Reviewer #2 (Remarks to the Author):

I read through the response letter and checked the revised manuscript. All the questions I had were properly addressed. I recommend this revised manuscript to be considered for publication.

Jinghang Xie

Reviewer #3 (Remarks to the Author):

Authors have completely addressed all the issues in this study. Therefore, I recommend the acceptance of the manuscript at current edition.

The following are our responses to the comments (*in Italics*) of Reviewer #1 and the changes (underlined) made in this manuscript (NCOMMS-18-33848A).

Reviewer #1

[1] *I agree the method for synthesizing the conjugated polymer is novel and the conjugated polymer has never been applied for photothermal therapy, but “the first time” does not always mean “novel”. Therefore, more comparison among other conjugated polymers (i.e. PANI, PEDOT, Ppy, et al.) should be conducted to highlight the uniqueness of the conjugated polymer in terms of photothermal therapy.]*

We thank the reviewer for him/her pointing out such important issues. Compared with other conjugated polymers that commonly used for preparing PTT agents (*e.g.*, PANI, PEDOT:PSS, and PPy), the synthesis of our PPBBT is relatively faster and more facile. Meanwhile, the preparation of the PPBBT-loaded water-soluble nanoparticles NP_{PPBBT} is routine and PTT efficiency of NP_{PPBBT} nanoparticles on tumours is comparable to that of other modified conjugated polymers. We revised accordingly.

Revision:

Page 18, line 4 from the bottom: Compared with other conjugated polymers that commonly used for preparing PTT agents (*e.g.*, polyaniline (PANI), poly-(3,4-ethylenedioxythiophene):poly(4-styrenesulfonate) (PEDOT:PSS), and polypyrrole (PPy))⁴⁰, the synthesis of our PPBBT is relatively faster and more facile. Meanwhile, the preparation procedure of the PPBBT-loaded water-soluble nanoparticles NP_{PPBBT} is routine and PTT efficiency of our NP_{PPBBT} nanoparticles on tumours is comparable to that of other modified conjugated polymers.

40 Vines, J. B., Lim, D.-J. & Park, H. Contemporary Polymer-Based Nanoparticle Systems for Photothermal Therapy. *Polymers* **10**, 1357 (2018).

[2] *For the biodistribution assay, it might not be accurate to use the ex vivo imaging system to quantify the actual amounts of nanomaterials that have been accumulated in specific organs for the following reasons:*
a: The self-quenching effect of fluorophore when the local concentration reached the threshold.]

We thank the reviewer for his/her instructive suggestion. In 2012, Hyunah Cho and *et al.* (*Nanomed.-Nanotechnol. Biol. Med.*, 2012, 8, 228-236) reported that, the fluorescence intensity of 1,1'-dioctadecyltetramethyl indotricarbocyanine iodide (DiR, the analog of DiD)

incorporated in PEG-*b*-PCL micelles was close to the fluorescence of DiR after 100% release from PEG-*b*-PCL micelles if the ratio of DiR : PEG-*b*-PCL was below 1:100. This result suggested that there was no fluorescence quenching in the micelles if the ratio was below 1:100. In our study, the ratio of DiD to mPEG-*b*-PHEP was 1:400. Thus, we believe that the fluorescence of DiD would not be quenched inside NP_{PPBBT}/DiD nanoparticles.

[b: The thickness of the organ might affect the fluorescence intensity.

Therefore, it is highly recommended to measure the fluorescence of the homogenized tissue after dilutions.]

As we know, whole organ analysis is a standard approach to study biodistribution.

Nevertheless, in current revision, we measured the fluorescence of the homogenized tissue to quantify the content of DiD in each organ. As shown in current Supplementary Figure 25, the content of DiD in tumours reached to its maximum at 24 h then gradually decreased. We revised accordingly.

Origin:

Page 15, line 251: Both the biodistribution data in ~~Figure 4B~~ and the *ex vivo* fluorescence images in Supplementary ~~Figure S23~~ of the major organs consistently indicated that, ~~tumours~~ had the highest average DiD fluorescence among all the major organs studied and their fluorescence reached to its maximum at 24 h then gradually decreased.

Revision:

Page 14, line 13: Both the biodistribution data in Fig. 4b, and the *ex vivo* fluorescence images and analyses in Supplementary Figures 24 and 25 of the major organs consistently indicated that, tumours had the highest average DiD fluorescence among all the major organs studied and their fluorescence reached to its maximum at 24 h then gradually decreased.

Current Supplementary Figure 25:

Supplementary Figure 25. Quantification of the content of DiD from the homogenized organs. **a** Fitted calibration curve of average radiant efficiency vs. content of DiD. **b** Quantification of the content of DiD (normalized by the tissue weight) from the homogenized organs (heart, liver, spleen, lung, kidney, and tumors) from Supplementary Figure 24. All organs were weighted and then homogenized in 1 mL 5% Triton X-100, respectively. Chloroform were used to extract DiD from homogenates for three times. The extracted DiD was redissolved in 500 μ L methanol after the chloroform was evaporated, and the obtained solutions were sent for fluorescent quantification using Xenogen IVIS® spectrum system. The samples of the standard curve were treated in the same way of the organs.